# Accelerating fragment-based drug discovery using grand canonical nonequilibrium candidate Monte Carlo

William G. Poole [1,2], Marley L. Samways [1,3], Davide Branduardi[2], Richard D. Taylor[3], Marcel L. Verdonk [2] & Jonathan W. Essex [1] ✉

Fragment-based drug discovery is a popular approach in the early stages of drug development. Computational tools are integral to these campaigns, providing a route to library design, virtual screening, the identification of putative small-molecule binding sites, the elucidation of binding geometries, and the prediction of accurate binding affinities. In this context, molecular dynamics-based simulations are increasingly popular, but often limited by sampling issues. Here, we develop grand canonical nonequilibrium candidate Monte Carlo (GCNCMC) to overcome these limitations. GCNCMC attempts the insertion and deletion of fragments to, or from, a region of interest; each proposed move is subject to a rigorous acceptance test based on the thermodynamic properties of the system. We demonstrate that fragment-based GCNCMC efficiently finds occluded fragment binding sites and accurately samples multiple binding modes. Finally, binding affinities of fragments are successfully calculated without the need for restraints, the handling of multiple binding modes, or symmetry corrections.

Fragment-based drug discovery (FBDD)[1] is becoming increasingly popular in industry to identify hits in the early stages of a drug discovery program, with 180 fragment-to-lead studies published between 2015 and 2022, and 7% of all clinical candidates published in the Journal of Medicinal Chemistry between 2018 and 2021 originating from a fragment screen[2–4]. Fundamental to FBDD is screening small libraries of diverse fragment molecules optimized to cover a large portion of chemical space. Typically, fragment libraries generate more, but less potent, hits compared to libraries of larger, more complex, molecules[1,5]. Although fragment hits tend to be weak binders, they generally provide a more useful starting point for hit-to-lead optimization compared to larger drug-like molecules owing to their simplicity, low molecular weight, rigidity, and numerous potential growth vectors[1,2,6]. Owing to their small size, fragments have very low binding affinities with dissociation constants ($K_D$) typically in the millimolar range[7]. Even with specialized biophysical techniques, these binding events are difficult, and sometimes impossible, to detect[5].

X-ray crystallography (XRC) can serve as hit detection while also providing a detailed structural picture of fragment binding[7,8]. However screening by XRC requires a reliable, high-throughput, and system-agnostic method of producing high-quality crystals of protein–ligand complexes. Unfortunately, for some protein targets (e.g. membrane and disordered proteins), producing such crystals can be difficult and costly. Furthermore, resolving the binding modes of small fragments can be tricky owing to unclear electron density, particularly for molecules with a high degree of symmetry or disorder[9–12]. Finally, XRC does not provide any information on how tightly a molecule is bound to a target, although one could, in principle, infer a limit for the binding affinity from the crystallization conditions[7]. Binding affinity data is desirable in drug discovery campaigns to rank hits based on how strongly they interact with a target, thus providing an additional critical component in deciding which fragments are the most promising for development.

[1]School of Chemistry and Chemical Engineering, University of Southampton, Southampton SO17 1BJ, UK. [2]Astex Pharmaceuticals, 436 Cambridge Science Park, Milton Road, Cambridge CB4 0QA, UK. [3]UCB, 216 Bath Road, Slough SL1 3WE, UK. ✉e-mail: j.w.essex@soton.ac.uk

Computational methods are well-placed to enhance the fragment-based drug discovery pipeline and complement experimental methods. A wealth of in silico methods are routinely used to identify binding sites, fragment hits, and to optimize hits to leads[6]. Even wider adoption of computational methods in FBDD is expected as they become more efficient and accurate, further accelerating drug discovery processes[6,13–15].

Docking is a computationally inexpensive technique that can be used to quickly generate a large variety of ligand-bound configurations and to rank these poses based on a scoring function[16]. However, many docking algorithms neglect protein dynamics, and flexibility, and often yield many false positives owing to inadequate scoring functions[13,16–18]. Moreover, it has been shown that many docking algorithms are optimized for larger, more drug-like molecules and perform much better when using crystal structures with bound analogues[19].

Molecular dynamics-based simulations explicitly capture protein dynamics but come with their own set of limitations. Spontaneous binding events from the solvent often occur over longer timescales (milliseconds) than can be simulated in a reasonable time frame (microseconds) and are therefore rarely observed[20–22]. Ligands that bind in multiple configurations can become restricted to only one, or two, modes with large energy barriers that prevent exchange between binding modes within the timescale of a typical MD simulation[23,24]. Many recent enhanced sampling methods have been developed to tackle such issues and have been applied in the context of FBDD[25–28].

Mixed solvent MD (MSMD) is a flavor of classical MD simulation that aims to identify binding sites and favorable interactions in protein systems using small organic probe molecules[14]. While MSMD has had some success in identifying binding sites and hotspots[14], it is plagued by time scale issues, particularly if the binding event requires a large protein conformational change, or if the binding site is particularly occluded from the solvent[14,27,29,30]. The choice of organic probe adds another level of complexity; ideally a range of probes would be selected to cover a variety of interaction types, however, some probes can aggregate, leading to the formation of clusters or even phase separation, limiting sampling at the protein interface by reducing the effective concentration of the system[31,32]. Some protocols make use of artificial repulsion forces between ligand pairs to maintain a well-mixed solution in simulation; however, this introduces artifacts and in theory could prevent cooperative binding[31,33]. Others use fully miscible probes, thereby avoiding the issue altogether, but this limits the probe set available and thus restricts the chemical space that can be sampled[29].

Similarly to XRC, structure-based computational methods also generate information on how a ligand binds in terms of its atomic positions. Ligands that bind in multiple configurations often produce ambiguous electron densities, and simulation methods can help to resolve these binding modes and generate an ensemble of stable configurations. Binding Modes of Ligands Using Enhanced Sampling (BLUES)[28] is one such method that aims to identify different stable binding modes within a binding site. Similar to the present study, BLUES uses nonequilibrium candidate Monte Carlo (NCMC) to fully decouple a bound ligand from a binding site before applying random translational, rotational, and dihedral adjustments to the ligand, followed by recoupling. While some degree of prior knowledge of the binding site is required, BLUES has successfully identified multiple known binding modes of fragments in T4-lysozyme and soluble epoxide hydrolase[28,34].

Finally, when a binding mode is known, alchemical free energy calculations[35–38] may be used to predict the affinity of the molecule to its target. Affinity prediction is very important and, when performed correctly, is an accurate and reliable way to rank a series of ligands[39,40]. Relative binding free energy (RBFE) calculations provide a means of predicting the free energy difference between two closely related ligands by perturbing one or two functional groups into another.

Absolute binding free energy (ABFE) calculations, on the other hand, calculate the overall binding affinity of a molecule to a target and are generally more useful when ranking ligands with different scaffolds; a retrospective study by Alibay et al. shows the applicability of this method in the context of FBDD[41]. However, for both methods, there is a requirement for high-quality structural data from either experiment or computation, prior knowledge of the binding modes, and, in the case of ABFEs, a series of user-defined restraints to maintain the integrity of the complex as the ligand is decoupled[42,43]. These restraints may require a degree of user input and the wrong choice can noticeably affect simulation convergence and even lead to simulation crashes[41,44]. In our experience, weakly bound fragments, that may try to unbind or swap biding modes, even in fully interacting states, test the robustness of these restraint protocols. Automated methods to generate ABFE restraint parameters exist[41,44–49] and these usually depend on running short preliminary simulations to identify stable restraint atoms. For highly mobile ligands the identification of a stable restraint may be difficult for the reasons outlined. Finally, sampling limitations, including the movement of protein side chains and tightly bound water molecules, have continued to be the source of further development[25,26,38,50,51].

Grand Canonical Monte Carlo (GCMC) simulations have been routinely used in recent years to simulate the grand canonical ($\mu$VT) ensemble, allowing the number of molecules in the system to fluctuate while keeping the overall chemical potential ($\mu$) of the system constant[52–55]. In practice, GCMC uses trial 'insertion' and 'deletion' moves to vary the number of molecules in the system; these trial moves are then subjected to a Monte Carlo test and are accepted or rejected according to the equilibrium properties of the system. Various applications of GCMC include sampling buried water molecules in protein-ligand binding regions, to validate crystal water positions, predict favorable water sites, and calculate the free energies of water networks[30,32,53,55–61]. Furthermore, water-based GCMC simulations are now also commonplace in free energy calculations, such as in the popular FEP+ software, as a means of improving water sampling while an alchemical change is applied to a molecule of interest[51,62].

A more recent study investigated the application of nonequilibrium candidate Monte Carlo (NCMC) to GCMC. In the context of GCMC, the addition of NCMC means that the insertion or deletion of molecules can occur gradually over a series of alchemical states such that the molecule binds with an induced fit mechanism allowing the system to respond to the proposed moves. When applied to water, the acceptance rate of this combined "Grand Canonical nonequilibrium candidate Monte Carlo" (GCNCMC) method was found to be significantly higher than 'instantaneous' GCMC, while also indirectly improving protein and ligand sampling during the move proposal[63,64]. Implementing GCNCMC moves into a regular MD simulation means that sampling of these binding sites is improved while also propagating the dynamics of the system through time.

Until now, our overarching goal of GCMC and GCNCMC has been to sample the binding and unbinding of tightly bound water molecules in buried protein regions where regular MD may otherwise struggle within a reasonable time scale. In much the same vein, simulations of ligand binding are also hampered by the relative time scales over which ligands exchange between a protein and solvent. To this end, we have further extended GCNCMC to sample the binding of small molecules to proteins where the induced fit mechanism is even more crucial. Note that by default we use the combined GCNCMC method for the sampling of small molecules as the acceptance rates of instantaneous GCMC moves are prohibitively low. We aim to demonstrate the application of GCNCMC to FBDD, addressing three of the aforementioned goals of in silico structure-based drug design: finding potential binding sites, identifying fragment binding modes, and predicting binding affinities. Conceptually, the method is almost identical to GCNCMC water sampling with few modifications, and for more

**1** Overcomes timescale limitations by rigorously inserting and deleting molecules into/from a specified region e.g. sphere

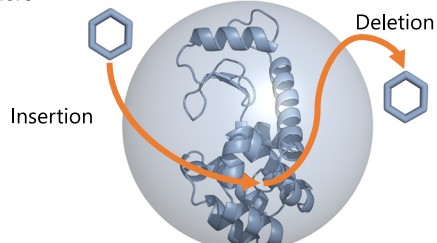

Deletion

Insertion

**3** Interspersing regular MD with GCNCMC moves allows the system to propagate through time

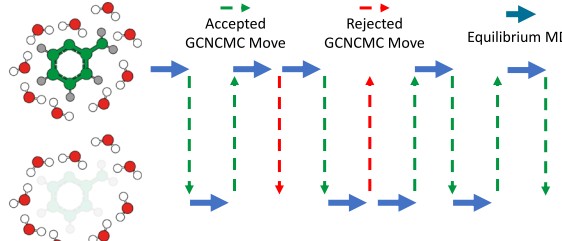

**2** Monte Carlo moves occur over a short, nonequilibrium, simulation via alchemical states. Moves accepted/rejected based on the thermodynamic properties of the system, including concentration and excess chemical potential

$\lambda=1.00$   $\lambda=0.75$   $\lambda=0.50$   $\lambda=0.25$   $\lambda=0.00$

$$\text{Insert} = \min\left[1, \frac{1}{N+1}e^{B_{eq}}e^{-\beta w(X|\Lambda_p)}\right]$$

$$\text{Deletion} = \min\left[1, N e^{-B_{eq}}e^{-\beta w(X|\Lambda_p)}\right]$$

**4** Simulating at multiple Adams values ($B_{eq}(c_L)$), and thereby concentrations, a free energy can be calculated via titration

$$B_{eq}(c_L) = \beta\mu'_{sol} + \ln(N_A c_L V_{GCMC})$$

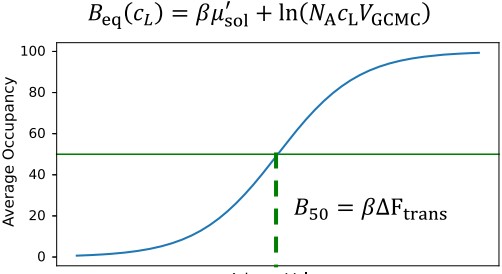

$B_{50} = \beta\Delta F_{trans}$

**Fig. 1 | High level overview of the GCNCMC protocol.** (1) Insertion and deletion moves occur within a user-defined region (gray sphere). (2) Moves are performed using a nonequilibrium switch occurring over a short time scale and are accepted or rejected according to the work done on the system over the move, $w(X|\Lambda_p)$, the excess chemical potential of the molecule, $\mu'_{sol}$, the concentration of the molecule, $c_L$, the number of molecules already in the region, $N$, and the volume of the defined GCMC region, $V_{GCMC}$. (3) The resulting simulation is a regular MD simulation with GCNCMC moves interspersed. If a move is rejected (red dashed lines) the simulation restarts from the state prior to the move. (4) Binding affinities may be calculated by titrating the Adams value, $B_{eq}$, and thereby concentration. More details can be found in the "Methods" section.

details, we refer the reader to the "Methods" section, and Supplementary Information, where the theoretical basis is outlined in depth.

Figure 1 gives a high-level overview of the GCNCMC protocol. In summary, the entire protocol involves running regular MD to propagate the system, with GCNCMC moves interspersed. An insertion or deletion move is selected with equal probability, and, for an insertion move, a non-interacting molecule is randomly placed into a user-defined GCMC region and 'switched on'. For a deletion move, a fully interacting molecule within the GCMC region is selected randomly and then 'switched off'. These switches are governed by a series of alchemical perturbations and a small amount of MD between each perturbation such that the entirety of the switch is performed out of equilibrium, or in other words, the simulation does not have time to equilibrate during the move. At the end of the GCNCMC move, a Monte Carlo acceptance test is performed and the move is either accepted or rejected based on the thermodynamic properties of the system including the excess chemical potential, and the desired concentration of the molecule being switched. Conceptually, higher concentrations would drive the acceptance of insertion moves and vice versa for deletion. If the move is accepted, the simulation continues from this state, and, if the move is rejected, the simulation restarts from the state prior to the attempted move.

In this work, we present the application of GCNCMC to the simulation of fragment-protein binding. We first provide a validation, by verifying that GCNCMC simulations accurately reproduce a simple ensemble property - bulk concentration of a solute in water. This is critical in ensuring the method, and its implementation, are working as intended. Second, we apply the method alongside MSMD simulations to enhance the binding of organic probe molecules in occluded binding pockets. We study two systems with occluded binding pockets, T4-lysozyme (T4L99A) and major urinary protein 1 (MUP1). Simulation studies of the former show that the binding of benzene

requires multi-microsecond long simulations which are unfeasible in a real world setting[20]. Using GCNCMC, we can overcome these time scale limitations by inserting molecules directly into the binding site; we can do this in a prospective manner by assuming that benzene could bind anywhere on the protein and setting the region in which GCNCMC moves can occur accordingly. We compare the sampling between our GCNCMC-enhanced MSMD simulations to classical MSMD simulations. We then focus our sampling on just binding sites, where we first establish whether GCNCMC can sample multiple binding modes correctly. Owing to the rigorous acceptance criteria for the GCNCMC moves, only the move proposals that generate a plausible configuration for a given concentration will be accepted. Through constant insertion and deletion move proposals, the resulting binding modes should be reproduced with appropriate populations. To confirm, we look at the binding modes in simulations of a model host-guest system ($\beta$-cyclodextrin) and for toluene binding to T4L99A. The latter has been shown to bind in two distinct orientations along with their symmetric equivalents, giving a total of four binding modes[28]. Finally, we then exploit the concentration dependence in the acceptance criteria to apply a titration protocol (Fig. 1) to calculate the binding affinities for small fragment molecules in the aforementioned systems. This method, unlike ABFE, does not require prior knowledge of binding modes, artificial restraints, and symmetry corrections. The results are compared to a more traditional ABFE calculation, which is often regarded as the gold standard in the field of computational free energy prediction. The protocols used are described in both the "Methods" section and the Supplementary Information. Thus, in this Communication, we show that GCNCMC: (1) can be used to accurately identify fragment binding sites within a protein structure; (2) identify multiple binding modes with correct populations without any a priori knowledge; (3) predict binding affinities which are in excellent agreement with more complex alchemical free energy methods.

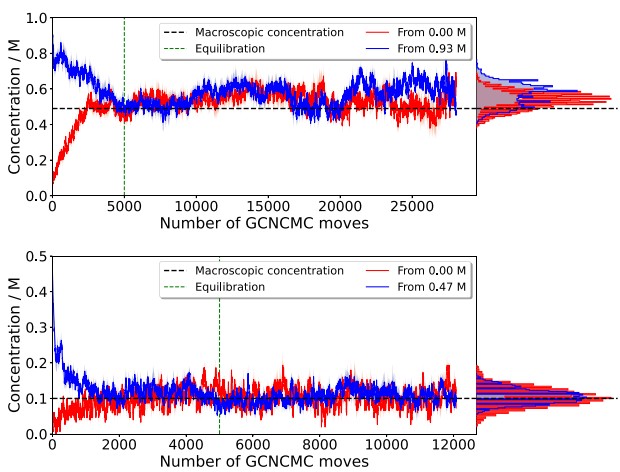

**Fig. 2 | Fragment concentrations as a function of time.** Top: GCNCMC simulation concentrations of acetone starting from a pure water box and a 1 M solution. Bottom: GCNCMC simulation concentrations of pyrimidine starting from a pure water box and a 0.5 M solution. Data points represent the mean concentration at each step over 8 (acetone) or 4 (pyrimidine) repeats. The shaded regions represent the standard error of the mean. Histograms are binned mean concentrations from after the equilibration point, indicated by the green dashed line. Equilibration was judged by eye as the point where systematic changes in the observed concentration were no longer apparent. Source data are provided as a Source Data file.

## Results and discussion

### GCNCMC simulations accurately reflect fragment concentrations in water

In previous work, we have validated our GCMC and GCNCMC methods by reproducing the mass density of TIP3P water boxes[59,63]. For fragment–water mixtures it is more appropriate to measure the bulk concentration of each fragment in water. We selected solutions of 0.5 M acetone and 0.1 M pyrimidine for this test as they do not aggregate in water at these concentrations. The starting concentrations for these tests were equilibrated boxes of pure water containing no other species, and boxes containing solutions of 1 M acetone and 0.5 M pyrimidine. To perform these simulations, we set the GCNCMC parameters appropriate for the target concentrations (see Methods).

Figure 2 shows the variation in concentration over simulation time for both fragments. In each case, after an appropriate equilibration period, the concentration fluctuates around the target value, demonstrating that not only can GCNCMC simulations maintain a defined concentration but also rapidly equilibrate the system. The final mean concentration for acetone was 0.55 ± 0.02 M and 0.56 ± 0.02 M when starting from 0 M and 1 M, respectively. While slightly higher than the desired concentration of 0.5 M, the consistency between the two systems is reassuring and indicates that the values of the excess chemical potential of either the ligand or water may not be sufficiently accurate to sample 0.5 M exactly. In the case of pyrimidine, a lower concentration of 0.1 M was selected and well reproduced. Further discussion of these results is available in the Supplementary Information.

### GCNCMC rapidly and accurately identifies occluded binding sites

A popular in silico method of finding potentially druggable sites is through MSMD simulations, whereby a protein is solvated in water with a high concentration of organic probe molecules[14]. Regions where probes are observed to spend a large amount of simulation time are deemed potentially druggable. These simulations generally work well for protein binding sites that are solvent exposed or are located on the surface of proteins, but for binding sites occluded from the solvent or more deeply buried, the simulation times required to overcome conformational energy barriers exceed what can be conveniently simulated[30,33].

Using GCNCMC, the limitations associated with diffusion times can be negated by performing insertion and deletion moves into or from the immediate vicinity of the protein. Setting the GCMC region to cover the whole, or parts, of the protein means that sampling can be focused only on these areas. This protocol is essentially a traditional MSMD simulation with enhanced sampling by GCNCMC. Here, we have applied this approach to find the occluded binding pockets of T4L99A and MUP1. To do this, we set our GCMC region to cover the whole protein by anchoring it to a central protein residue.

T4L99A is an extensively studied test system with a wealth of experimental data, commonly used in the development of free energy methods[20,65–69]. A point mutation (L99A) results in a protein cavity which binds a range of small aromatic ligands. The T4L99A system, while relatively simple, does have some complexities that make it particularly interesting for the testing of enhanced sampling methods. For example, as the binding site is completely occluded from the solvent, MSMD simulations struggle to map the site using classical MD owing to the timescales required for diffusion[30,33,70]. Secondly, some ligands, such as toluene, bind to T4L99A in multiple orientations with a significant kinetic barrier preventing them from readily interchanging[28]. Here, using benzene as a probe, we first test GCNCMC as a site finding tool and compare its ability to map the occluded binding pocket to classical MSMD simulations.

Across our six repeats, simulated at 0.5 M probe concentration, the benzene binding site was readily found within an average of 34 GCNCMC moves. For context, we ran 700 moves per repeat in 24 hours of wall time on a GTX1080 GPU. Once the site was found, the ligand remained for the rest of the simulation as every deletion proposal was deemed unfavorable at this concentration. By binning the coordinates of sampled benzene heavy atoms onto a grid with 0.5 Å resolution, we can count the number of frames a benzene atom was present at each grid point and then average based on the total number of frames. By contouring this grid to represent an occupancy of at least 90% of our frames (Fig. 3), we see a clear signal around the crystallographic binding pose indicating that a benzene molecule was present in this site for at least 90% of our simulation. Further, as no other grid points are occupied at such high percentages, the binding site is clear with a lack of false positives.

These results are compared to a basic MSMD simulation of T4L99A in a 0.5 M solution of benzene and are shown in Fig. 3. The MD grid is contoured at 30% indicating the grid points where a benzene atom had resided for 30% or more of the simulation frames, showing that the benzene binding site was not sampled at all. Turning this contour level down to 1% still reveals no binding. This grid-based analysis could be exploited to give a rough estimate of free energy by comparing its occupancy to that of bulk solvent, in a fashion similar to other MSMD methods[14].

Various computational studies of benzene binding to T4L99A, including long MD simulations, have shown that the binding of benzene can take tens of microseconds[70]. In an industrial research and development setting, simulations of this time scale are impractical and expensive, further highlighting the benefits of the GCNCMC approach, namely its efficiency: An average of 34 moves at 150 ps switching time is equivalent to approximately 5 ns of simulation time (accounting for the MD steps between each GCNCMC trial).

Like T4L99A, MUP1 is another protein system with an occluded binding pocket and has been used as a test system for relative binding free energy calculations[37,71]. Again, we perform GCNCMC enhanced MSMD simulations using three different fragment molecules that are known to bind to MUP1, **07, 08**, and **14** Supplementary

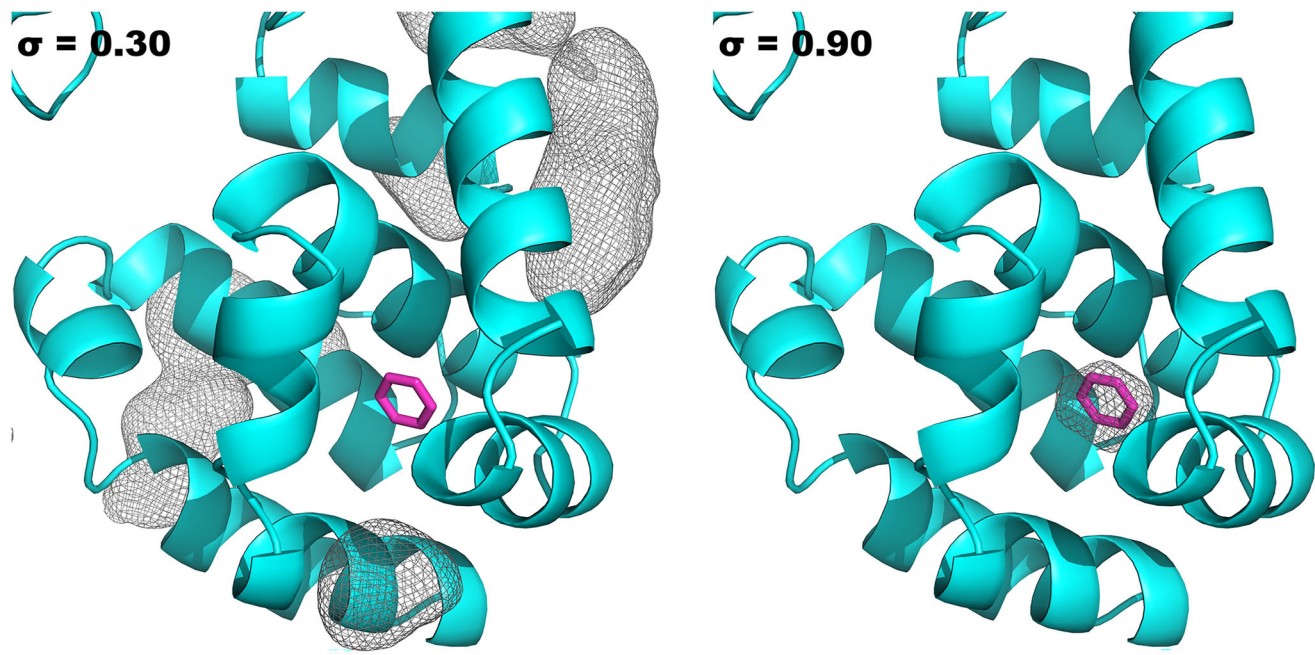

**Fig. 3 | Occupancy grids of MD (left) and GCNCMC simulations (right) contoured at a value of 0.30 and 0.90, respectively.** Grids represent a minimum of 30% and 90% of the frames for which a benzene atom visited a given grid point. The benzene crystal pose is shown in magenta (PDB: 181l).

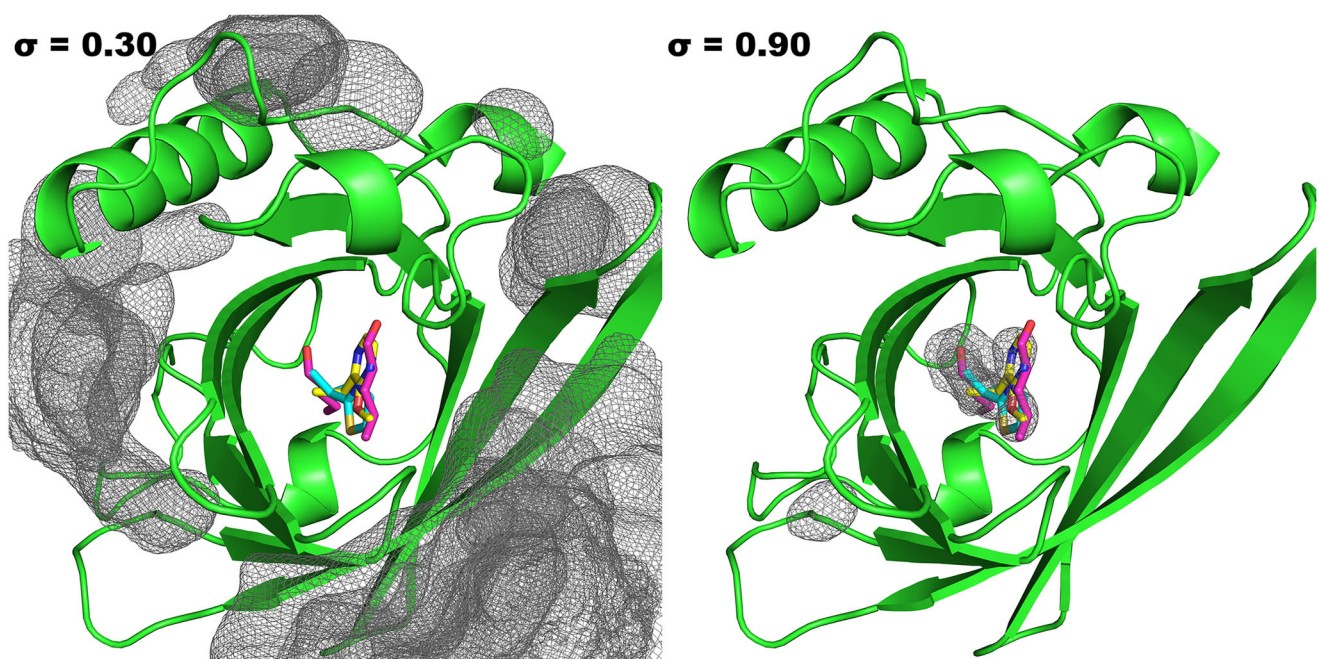

**Fig. 4 | Occupancy grids of MD (left) and GCNCMC simulations (right) contoured at a value of 0.30 and 0.90, respectively.** Grids represent a minimum of 30% and 90% of the frames for which a grid point was occupied by a ligand. The grids for all three MUP1 ligands (**07, 08**, and **14**) are shown together. Representative crystal structures for each ligand are shown in cyan (PDB: 1i06), magenta (1znd), and yellow (1qy2).

Fig. 23. As for T4L99A, we are attempting to predict the known binding sites starting from an unbound structure. In Fig. 4 the MD and GCNCMC occupancy grids are presented. Immediately, it is clear that basic MD fails to sample the binding pocket, and even setting the contour level of the MD occupancy grids to 1% still shows no binding. As for T4L99A, this lack of binding can be attributed to two factors: first, the binding site is completely occluded from the bulk solvent and, second, at 0.5 M, all three fragments aggregated, severely impacting the level of sampling that is achievable in the simulation.

These results have positive implications for structure-based drug design in locating binding pockets for protein targets that may not be amenable to x-ray crystallography. Our method enhances traditional MSMD approaches which struggle to find occluded pockets. We will expand this simple use case of GCNCMC in the future.

### GCNCMC automatically captures multiple binding modes without prior knowledge

Once a binding site is known, it is useful to understand all the possible ways in which a fragment can bind in that site. Experimental data,

## Primary Opening

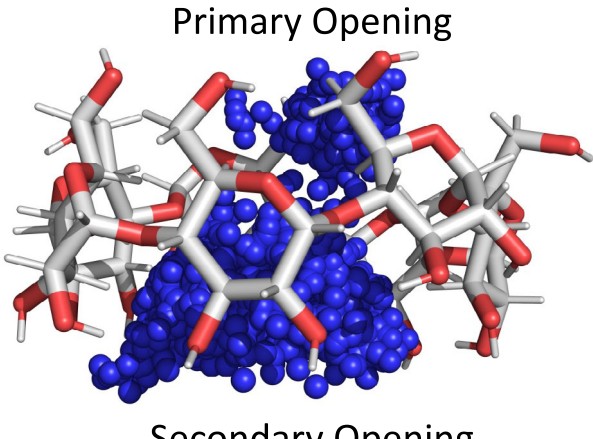

## Secondary Opening

**Fig. 5 | Overlaid frames from GCNCMC simulations of benzonitrile binding to β-cyclodextrin.** GCNCMC simulations show a preference for the polar group of the benzonitrile guest (blue spheres) to point out the wider opening composed of secondary alcohols. Note, that the depiction of the host is that of the first frame only.

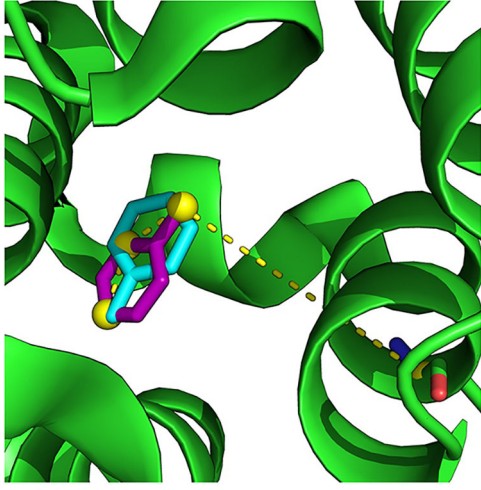

**Fig. 6 | The two binding modes of Toluene to T4L99A.** The crystal pose, A1/A2, is shown in magenta (PDB: 4w53) and the meta-stable pose, B1/B2, is in cyan. The dihedral angle between the three Toluene atoms highlighted in yellow, and the CA atom of Arg119 is used to distinguish between the binding modes.

including crystal structures and binding affinities, are ensemble averages over all these possible configurations. However, in MD simulations, the kinetic barrier associated with changing binding mode can be very high making this process challenging to sample. If medicinal chemists do not fully understand how a fragment binds, crucial information on how to optimize or exploit certain binding interactions may be lost. Here, we explore the ability of GCNCMC to sample multiple binding modes without any prior knowledge of their existence.

Host-guest systems, such as β-cyclodextrin (βCD), are convenient and tractable complexes that are often used to test new simulation methods, and notably exhibit many of the same characteristics as fragment binding to proteins. Owing to their small size, simulations of host-guest systems generally converge quickly[72–76].

It has been reported that guests with a single polar group bind to βCD in two distinct orientations with the polar group pointing out of each end of the host, with ligands binding more favorably in the secondary alcohol orientation (Fig. 5)[72]. To check if these binding modes are observed in our GCNCMC simulations, we pull frames from titration studies (Section "GCNCMC titrations can accurately rank binding affinities") for two representative fragments, benzonitrile (Fig. 5) and para-cresol (Supplementary Fig. 10). Specifically, we look at simulations with B values that gave approximately a 50% occupancy ($B_{50}$), as these are the B values corresponding to the dissociation constant, $K_D$ (see "Methods"), and result in the maximum number of binding and unbinding events. We overlay these frames and see that the polar group for both ligands, as expected, preferentially points out the wider secondary alcohol opening of βCD (Fig. 5).

In a more quantitative study of multiple binding modes, we look at toluene binding to T4L99A, which has previously published simulation data from Gill et al.[28] In that study, two toluene binding poses were identified: the crystal pose (denoted A1/A2) and a secondary pose (B1/B2). Our own absolute binding free energy calculations of both poses revealed a free energy difference of 0.79 $k_BT$, which translates to a population ratio of 69:31 at 298 K for poses A and B, respectively (Fig. 6). This is in line with the ratio of 65:35 at 300 K reported by Gill et al.[28] Note that A1/A2 corresponds to the symmetrically equivalent poses of A and likewise for B, owing to the $C_2$ symmetry axis of toluene.

Using the dihedral angle identified by Gill et al. to distinguish between the bound configurations A and B, the populations obtained from GCNCMC simulations are shown in Fig. 7. We observed a ratio of

67:33 between poses A and B, which is in remarkably good agreement with our free energy estimates (69:31) and that of 65:35 reported by Gill et al. Reassuringly, we also observe population ratios of 33:34 between symmetry-equivalent poses A1/A2, and 17:16 between B1/B2, indicating thorough sampling. An interesting distinction between these results and those published by Gill et al. is that when using GCNCMC, we did not have to define a transformation between the poses (taken as center-of-mass rotation by Gill et al. when using BLUES[28]). Further, with GCNCMC, no prior knowledge of the multiple binding modes is required, as we have shown that these are naturally sampled by the method giving accurate populations.

In an alternative analysis, using CLonE[77], we have clustered the pairwise RMSD of the ligand positions throughout the simulation. The data are then projected onto a latent space using Principal Component Analysis. The populations of the four binding modes are in good agreement with those obtained using the dihedral histograms (Fig. 7, inset). This analysis is more general, as knowledge of a dihedral angle that discriminates between the binding mode is not required. This clustering analysis is used later to identify multiple binding modes in GCNCMC titrations Supplementary Fig. 14.

These results provide further validation that all binding modes, and their symmetry equivalents, are sampled within a GCNCMC simulation. Consequently, this means that binding modes are inherently sampled in our titration calculations (Section "GCNCMC Titrations can Accurately Rank Binding Affinities"), and thus are accounted for in the final free energy estimates, meaning there is no need for prior knowledge of the binding modes nor separate calculations for each mode.

### GCNCMC titrations can accurately rank binding affinities

Finally, now that we have validated the ability of GCNCMC, without prior knowledge, to predict binding sites and relative binding mode populations in a given site, we can apply our titration protocol to calculate the binding affinities of different fragment series to β-cyclodextrin, T4L99A, and MUP1. The structures of the fragments are shown in the Supplementary Information.

Using a pre-calculated excess chemical potential for each ligand, as would be necessary for any free energy calculation, we simulate over a range of ligand concentrations via Adams value scanning (see Methods section). We then measure the average occupancy of the simulation at each value of $B_{eq}$ and fit a logistic function, Eq. (18), to these data. The fragment concentration for which the corresponding

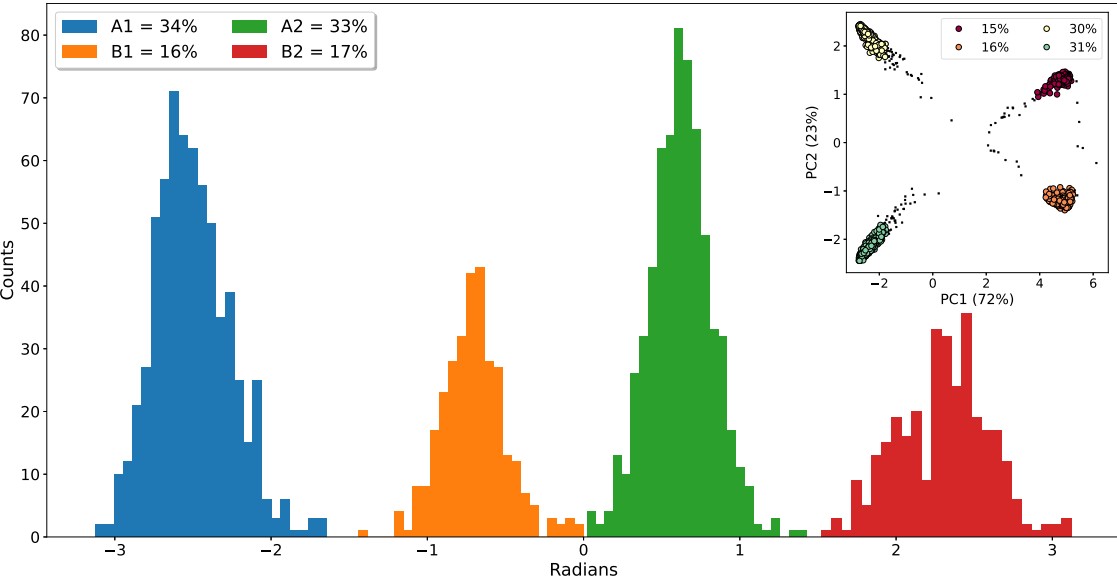

**Fig. 7 | Distribution of Toluene binding modes observed in GCNCMC/MD simulations.** Dihedral angles between $-\pi$ to $-1.5$ and 0 to 1.5 were assigned to binding modes A1 and A2, respectively. B1/B2 were assigned to angles between $-1.5$ to 0 and 1.5 to $\pi$. Inset: Pairwise RMSD between ligand poses projected onto PCA space and colored by the four clustered binding modes. Source data are provided as a Source Data file.

average occupancy is 50% is the dissociation constant, $K_D$, which is easily related to the free energy of binding (Eq. (13)).

Individual titration curves for 22 guests binding to $\beta$CD can be plotted together, giving a quick and easy indication of the strongest (far left) and weakest (far right) binding fragments (Fig. 8). These plots give valuable information concerning the binding process and are readily interpreted.

Figure 8 depicts the host-guest binding free energies extracted from the GCNCMC titrations, compared to experimental data and a basic ABFE approach which uses a flat bottom restraint to keep the guest bound as it is decoupled. In general, a slight overestimation of the binding affinities relative to experiment was observed, prompting speculation that the forcefield parameters used may not be optimal, with similar trends having been reported previously[72]. Despite this, the calculations gave a mean absolute error (MAE) and root mean squared error (RMSE) with respect to the experiment of 0.7 and 0.6 kcal mol$^{-1}$, with almost all the data points falling within 1 kcal mol$^{-1}$ of the experimental value. Furthermore, the correlation ($R^2 = 0.94$) and ranking ($\tau = 0.84$) with respect to experiment shows that the method can reliably and accurately rank fragments in terms of their binding affinities. An almost perfect correlation with the more well-established FEP approach gives promising validation.

Titration curves for the two protein systems, T4L99A and MUP1, can be found in the Supplementary Information. Figure 9 shows the calculated affinities versus experimental data and ABFE calculations. For T4L99A, the correlation with experiment ($R^2 = 0.562$) is comparable to other methods[66] and on par with our FEP protocol ($R^2 = 0.552$, Supplementary Information). However, the average error and RMSE with respect to experiment are particularly high, highlighting the added complexity of a protein system compared to a simple host-guest test case. Phenol and 2-Fluorobenzaldehyde were included as negative controls. GCNCMC titrations of Phenol, which is thought to predominantly bind to the unfolded state of T4L99A, predicted a slightly negative binding free energy ($K_D = 0.21M$). It is possible that this weak binding is masked in the thermal shift assay by preferential binding to the unfolded protein[78]. Crucially, the titration results are in good agreement with those obtained using FEP ($R^2 = 0.812$) which implies that the simulation methods are consistent. As with the host–guest system, GCNCMC titrations effectively sample the multiple

binding modes of T4L99A ligands as highlighted in Supplementary Fig. 14.

Finally, titrations of MUP1, which binds a diverse set of fragments, returned a good correlation with experimental data and an almost perfect correlation with FEP results. Interestingly, some of the fragments that bind to MUP1, such as octanol, have many degrees of freedom, and it is reassuring to see these sampled sufficiently.

Crucial to these results is that, as distinct and symmetrically equivalent binding modes are sampled naturally by GCNCMC, there is no need to perform more than one set of GCNCMC calculations, or to apply *post-hoc* symmetry corrections, as is required in ABFE methods[43,79]. Second, this method of calculating free energies does not require the use of artificial restraints, which can become problematic and may require some degree of user input[42,44]. However, it should be noted that when ligand binding induces a side chain movement, for example, the rotamer flip of Val111 upon binding of para-xylene to T4L99A, then to obtain accurate nonequilibrium work values, and thus free energies, this side chain movement must be sampled during the insertion and deletion moves. Unfortunately, in line with many similar methods, these side chain movements are difficult to sample, particularly over the timescales used in non-equilibrium switching[66,80,81]. Of course, it is possible to couple these side chain movements into the insertion and deletion move proposal but this would require prior knowledge of the system and modifications to the code. Alternative possibilities include sampling the valine dihedral using other enhanced sampling methods, such as BLUES[50] or FAST[26], in between GCNCMC moves. This case is discussed further in the Supplementary Information.

## Summary

In this Communication, we have further developed and applied our grand canonical nonequilibrium Monte Carlo method[63] to sample the binding of small molecules to protein systems with the goal of providing a tool which can help in the discovery of new fragment leads. The method has been successfully applied to detect binding sites, elucidate binding geometries, and calculate binding affinities.

We have validated the method by reproducing a simple ensemble property, namely fragment concentration in water. We showed that our GCNCMC implementation accurately recreates user-defined

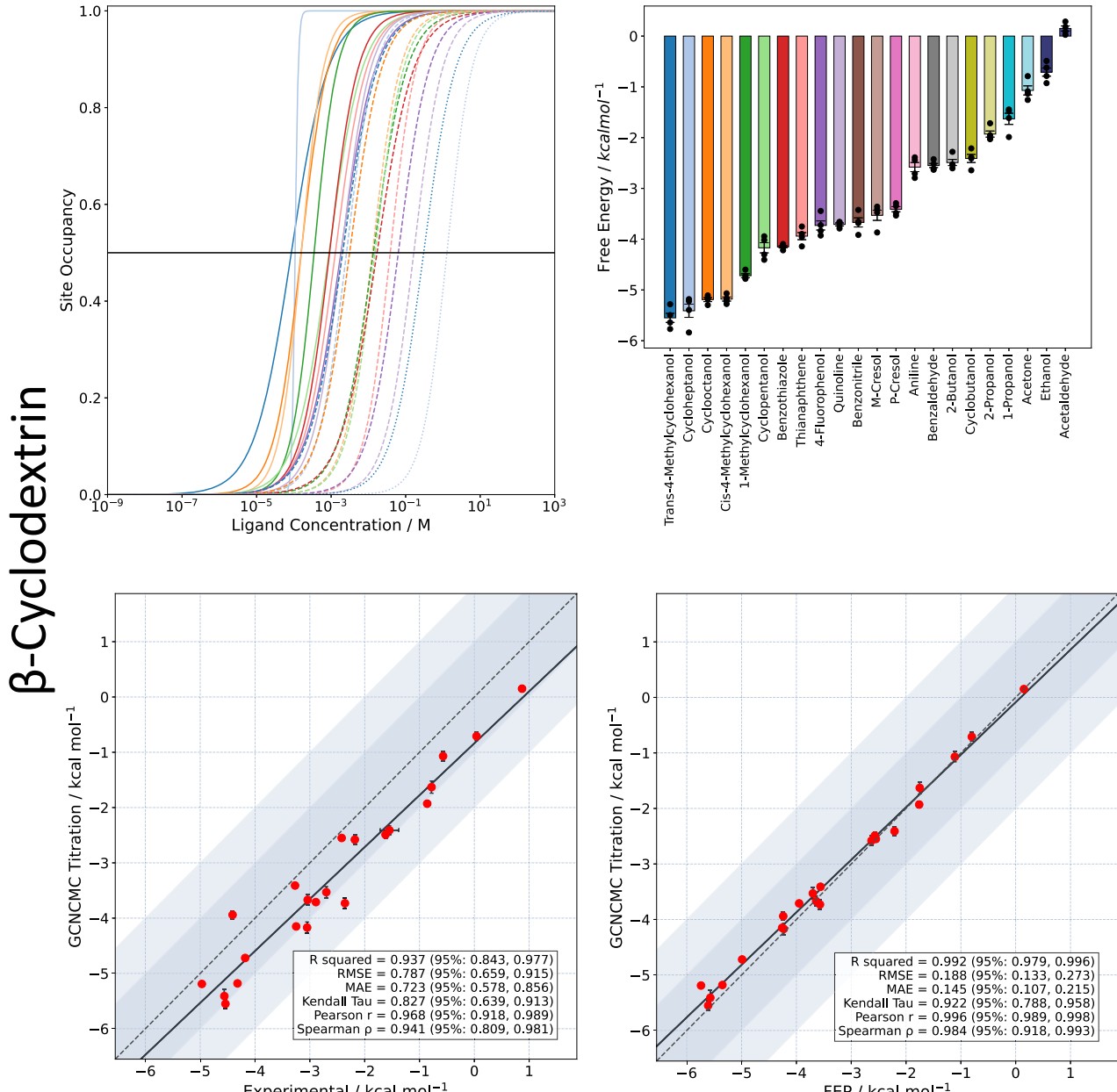

**Fig. 8 | Binding free energy data for the 22 tested fragment molecules binding to βCD.** Top: Titration curves (left) and binding free energy (right), the latter derived from the mean $K_D$ from four simulation repeats, each fitted to a sigmoid curve, and is reported in units of kcal mol⁻¹. The error is the standard error of the mean of the four $K_D$ values obtained from these fits. Bottom: Calculated absolute

binding free energies from titration calculations vs. experiment and FEP results, the latter obtained using a flat bottom restraint. The error on the ABFE results are the standard error of the mean of 4 individual repeats (2 starting from each binding mode). The shaded blue region symmetric around the y=x line represents 1-2 kcal mol⁻¹ deviations from the y=x line. Source data are provided as a Source Data file.

fragment concentrations, although, at high concentrations, the results become sensitive to the value of the excess chemical potential used.

We applied the method to two protein-fragment systems, T4L99A and MUP1, to enhance traditional MSMD simulations. In both cases, we assumed no prior knowledge of the binding region and subsequently ran GCNCMC simulations to generate a set of possible binding sites. We conducted a comparative analysis between GCNCMC and a basic MSMD protocol, revealing that GCNCMC readily identified experimental binding sites in both systems, whereas MSMD did not.

We then demonstrated that, as expected, the method reproduced fragment binding modes. We found that GCNCMC sampled the two

binding modes of simple fragments in a host–guest system with our simulations showing the guest molecules binding in an orientation in line with experimental and other reported data[72]. A more detailed examination of Toluene binding to T4L99A showed that the method successfully replicated the four binding modes with populations consistent with previously published studies[34].

GCNCMC titrations were then used to calculate the binding affinities of a series of small molecules to the host molecule, β-cyclodextrin, and the protein systems T4L99A and MUP1. We found that for the relatively simple βCD test case, a good correlation with experimental data, and an orthogonal computational method (ABFE calculations with restraints), was observed. This agreement with a well-established method provides solid

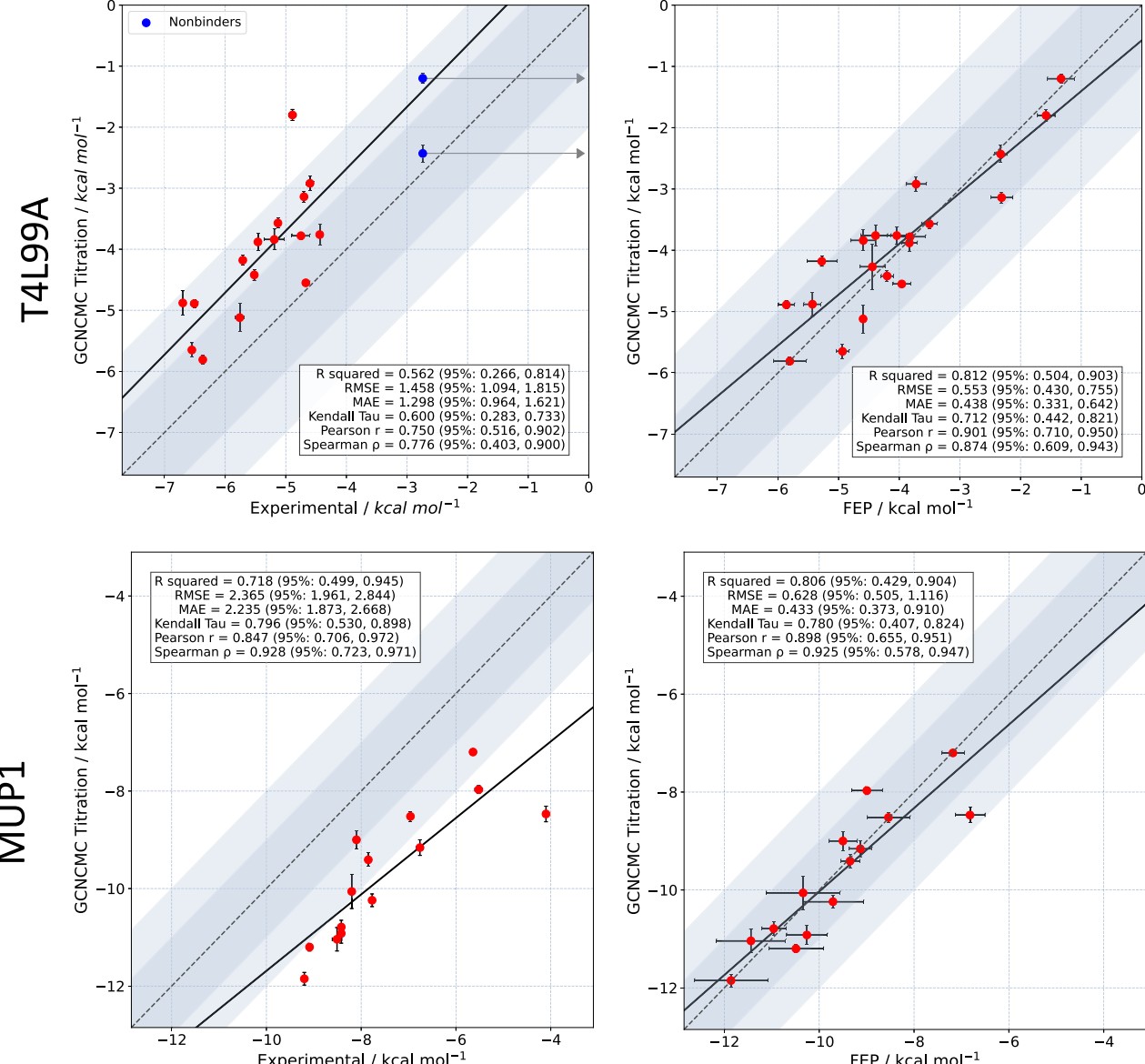

**Fig. 9 | Calculated binding free energies from titration calculations vs. experiment and FEP results, using Boresch restraints.** Top: T4L99A, Bottom: MUP1. Titration free energies are derived from the mean $K_D$ values of four simulation repeats, each fitted to a sigmoid curve, and are reported in units of kcal mol⁻¹. The error is the standard error of the mean of the four $K_D$ obtained from these fits. In both T4L99A and MUP1, ABFE calculations used appropriately weighted binding free energies derived from independent simulations of all populated binding geometries with a greater than 10% observed occupancy in GCNCMC titrations. The error on the ABFE results are the standard error of the mean of 3 individual repeats. For the comparison with experimental ligand binding free energies in T4L99A, data are only presented for compounds with experimental ITC data. Phenol and 2-Fluorobenzaldehyde are shown in blue and have a minimum experimental binding free energy of −2.74 kcal mol⁻¹. These compounds are not included in the reported statistics and line of best fit data. The shaded blue region symmetric around the $y = x$ line represents 1–2 kcal mol⁻¹ deviations from the y=x line. Source data are provided as a Source Data file.

validation and is important to demonstrate when developing a new free energy approach. Likewise, similar behavior was seen in both protein systems.

Owing to their small size, fragments have a propensity to bind in multiple orientations, making ABFE calculations in this context problematic. Given that GCNCMC samples binding modes naturally, affinity calculations using GCNCMC titrations eliminate the need for binding mode prediction, artificial restraints, and symmetry corrections, all of which are necessary in ABFE calculations, and may require some degree of user input. Crucially, unlike ABFE, there is no need to perform more than one set of calculations per molecule.

The initial goal of expanding the applicability of GCNCMC to small molecules was to improve the sampling of binding in occluded pockets, in much the same way as for water[63]. While this goal has been achieved, some caveats should be recognized. As noted here and elsewhere, alchemical free energy calculations are hampered by a number of sampling challenges[38,51]. Some of these issues - notably knowledge and treatment of multiple binding modes - are addressed by GCNCMC. However, other sampling challenges remain - in particular, when the alchemical transformation (in this case, fragment insertion/deletion) requires the concomitant binding or displacement of solvent molecules, or protein conformational change. For example, if a fragment

insertion move fails to displace bound water molecules, the move will likely either fail to be accepted at an appropriate concentration, or fail to sample the correct fragment binding mode. Future work will therefore seek to combine fragment GCNCMC with other enhanced sampling techniques to address these limitations.

This study offers robust validation and establishes a proof of concept for the use of fragment GCNCMC in the context of fragment-based drug discovery. Given GCNCMC's dual capability as a free energy estimator and a rapid and accurate pocket detection tool, fragment GCNCMC holds significant promise. As the method matures and evolves, we will apply it to more interesting and pharmaceutically relevant targets, particularly those that require protein conformational change and/or water displacement.

## Methods

### Theory

**Grand canonical nonequilibrium candidate Monte Carlo.** Ligand-based grand canonical nonequilibrium candidate Monte Carlo (GCNCMC) builds on the work of Samways et al. and Melling et al.[59,63,82–84] The final result is presented here with a full derivation in the Supplementary Information.

The acceptance probabilities of GCNCMC insertion and deletion moves occurring within a cubic system are given (using the Adams formulation)[52,63]:

$$P_{\text{insert}} = \min\left[1, \frac{1}{N+1} e^{B_{\text{eq}}} e^{-\beta w(X|\Lambda_{\text{p}})}\right] \quad (1)$$

$$P_{\text{delete}} = \min\left[1, N e^{-B_{\text{eq}}} e^{-\beta w(X|\Lambda_{\text{p}})}\right] \quad (2)$$

where $w(X|\Lambda_{\text{p}})$ is the work done in generating the trajectory $X$ by the protocol $\Lambda_{\text{p}}$, or in other words, the work done by the nonequilibrium switch, N is the number of molecules in the initial state immediately prior to the move, and $B_{\text{eq}}$ is the Adams value. For a particular fragment concentration $B_{\text{eq}}(c)$ is given by

$$B_{\text{eq}}(c) = \beta\mu'_{\text{sol}} + \ln\left(\frac{V_{\text{GCMC}}}{V(c)}\right) \quad (3)$$

where $\mu'_{\text{sol}}$ is the excess chemical potential of the ligand molecule which is usually approximated as the infinitely dilute standard solvation free energy of the molecule. $c$ is the concentration of the molecules in the reference solution with which the protein is in contact, $V(c)$ is the average volume per molecule in the solution at the specified concentration, and $V_{\text{GCMC}}$ is the volume of the GCMC region.

For a given reference concentration, the average volume per molecule can be trivially calculated as

$$V(c) = \frac{1}{cN_{\text{A}}} \quad (4)$$

where $N_{\text{A}}$ is Avogadro's constant.

Finally, the length of a nonequilibrium move is governed by the total number of perturbation steps, $n_{\text{pert}}$, between the two end states ($\lambda = 0$ and $\lambda = 1$), and the number of MD propagation steps between each perturbation, $n_{\text{prop}}$. Together this gives an equation for the switching time, $\tau$, as

$$\tau = (n_{\text{pert}} + 1)n_{\text{prop}}\delta t \quad (5)$$

where $\delta t$ is the integrator time step. To maintain detailed balance, each GCNCMC move starts and ends with a propagation step.

**The GCMC Sphere.** To improve sampling of a particular region of interest, the user can target GCNCMC moves by defining a spherical "GCMC region" to cover this region. For example, this could be a known binding site or the whole protein For example, this could be a known binding site or the whole protein (Supplementary Fig. 2). However, the use of a sphere for GCNCMC moves requires special care, in that molecules can potentially diffuse into or out of the sphere throughout the switch, meaning that the acceptance ratios defined previously must be adjusted:

$$P_{\text{insert}} = \min\left[1, \frac{1}{N_T} e^{B_{\text{eq}}} e^{-\beta w(X|\Lambda_{\text{p}})}\right] \quad (6)$$

$$P_{\text{delete}} = \min\left[1, N_0 e^{-B_{\text{eq}}} e^{-\beta w(X|\Lambda_{\text{p}})}\right] \quad (7)$$

where $N_0$ is the number of GCMC molecules in the sphere at the start of a move and $N_T$ is the number of molecules at the end. It should be noted that if the molecule being switched lies outside the sphere by the end of the move, the move must be automatically rejected since the reverse process cannot be proposed, breaking the condition for detailed balance. Depending on the characteristics of the binding site, this can sometimes lead to a high number of moves being rejected.

**The Adams Value.** In the previous section, we introduced the Adams value, $B_{\text{eq}}$. This is the controlling parameter in GCNCMC and ultimately determines whether a GCNCMC move is to be accepted or rejected. It therefore requires some care in its definitions.

In our previous publications of water GCMC and GCNCMC[59,63], the Adams value is defined as

$$B_{\text{eq}} = \beta\mu'_{\text{sol}} + \ln\left(\frac{V_{\text{GCMC}}}{V^{\ominus}}\right) \quad (8)$$

where $\mu'_{\text{sol}}$ is the excess chemical potential of the molecule and $V^{\ominus}$ is the standard state volume of the molecule of interest. The standard states for water and small molecules are well defined as 55 M and 1 M, respectively, however, in many cases, simulating a molecule, such as a fragment, at a concentration that is not the standard state is more experimentally relevant. For example, fragment-like molecules tend to bind to their targets in the micromolar to millimolar range. In situations where the molecule in a reference solution (the solution with which our simulated system is in equilibrium) deviates from the standard state, we define the Adams value with a specific concentration dependence:

$$B_{\text{eq}}(c) = \beta\mu'_{\text{sol}} + \ln\left(\frac{V_{\text{GCMC}}}{V(c)}\right) \quad (3)$$

where $V(c)$ is now the average volume occupied by a molecule at concentration, $c$. This equation can also be written in terms of concentration directly:

$$B_{\text{eq}} = \beta\mu'_{\text{sol}} + \ln(N_{\text{A}}c_L V_{\text{GCMC}}) \quad (9)$$

where

$$c_L = \frac{1}{N_{\text{A}}V(c)} \quad (10)$$

The excess chemical potential of a molecule is defined as the free energy of adding a molecule to a given solution and is crucial

in determining whether a given fragment would prefer to be in solution, or a binding site. For the binding studies this work, we approximate the excess chemical potential of a given fragment to be equal to the "infinite dilution" hydration free energy of that fragment, or in other words, the free energy of adding a fragment molecule to a box of water. However, it should be emphasized that the value of $\mu'_{sol}$ can also be influenced by interactions with other molecules of the same type and in theory also has a concentration dependence[85]. Nevertheless, for sufficiently dilute concentrations such as those used in these studies, it is assumed that the impact of these interactions is negligible and that $\mu'_{sol}$ is independent of concentration. As a result, for a given molecule, only one value of $\mu'_{sol}$ requires calculation, and this value can be applied to any dilute concentration of that molecule. A full discussion on this point, with data, can be found in the Supplementary Information.

With $\mu'_{sol}$ fixed, the Adams value is dependent only on the concentration of fragment in the reference solution. It is now intuitive to say that a higher reference solution concentration would lead to more binding in the GCMC region of the protein, as shown in Fig. 10. As explained in the next section and Supplementary Information, the concentration of fragment which leads to 50% occupancy in a binding site is equivalent to the dissociation constant, $K_D$. We define the corresponding Adams value as $B_{50}$:

$$B_{50}(K_D) = \beta\mu'_{sol} + \ln\left(\frac{V_{GCMC}}{V(K_D)}\right) \tag{11}$$

It is clear that simulations performed at $B_{50}$ will result in the maximum number of accepted insertion and deletion moves to maintain a 50% occupancy thus resulting in maximal binding and unbinding events.

**Free energies of binding from GCNCMC titrations.** To calculate binding affinities using GCNCMC we exploit the concentration dependence in $B_{eq}$ to perform titrations over a range of Adams values, $B_{eq}$, and thus the range of concentrations with which our GCMC region is in equilibrium[55,60]. The binding process of a ligand to a protein can be defined by the following equilibrium:

$$L + P \rightleftharpoons LP$$

where the equilibrium constant for the unbinding process, known as the dissociation constant, $K_D$, is given as a ratio of the concentrations of the species:

$$K_D = \frac{[L][P]}{c^\ominus[LP]} \tag{12}$$

where $[P]$, $[L]$ and $[LP]$ are the molar concentrations of the protein, ligand, and complex, respectively, and $c^\ominus$ is the standard state concentration, taken to be 1M by convention for a ligand in solution. In the simple, and most common case, of one ligand binding in one binding site, we can calculate $K_D$ as the dimensionless ligand concentration, $\frac{[L]}{c^\ominus}$, at the point at which the concentration of the bound protein is equal to the concentration of the free protein, $[LP] = [P]$.

This corresponds to the concentration of the ligand that binds half of the receptor and manifests itself in a GCNCMC simulation as the ligand concentration required so that the receptor is bound for half of the simulation (50% occupancy). This is also the concentration that gives equal acceptance probabilities for both insertion and deletion moves, resulting in maximal binding and unbinding. We refer to the corresponding Adams values as $B_{50}$ throughout.

Given the concentration dependence of $B_{eq}$, it follows that by titrating the Adams value, binding affinities can be calculated by finding the ligand concentration at which 50% of the protein is bound. This results in the ligand dissociation constant, $K_D$, and is then easily related to the standard Gibbs free energy change of binding using Eq. (13)[55,60].

$$\Delta G^\ominus = k_B T \ln K_D \tag{13}$$

A simple rearrangement of Eq. (3) and (4) shows how the B value can be related to a species' concentration in solution:

$$c_L = \frac{e^{B_{eq} - \beta\mu'_{sol}}}{N_A V_{GCMC}} \tag{14}$$

As mentioned previously, the value of $\mu'_{sol}$ here needs to be pre-calculated via a hydration free energy calculation of the ligand of interest.

In our previous work[55], it was shown that the binding of a single water molecule can be described by a logistic function in terms of the Adams value:

$$N(B_{eq}) = \frac{1}{1 + \exp[k(B_{50} - B_{eq})]} \tag{15}$$

where $N(B_{eq})$ is the average number of molecules in the GCMC region as a function of $B_{eq}$. $B_{50}$ is the Adams value required to return a 50% bound occupancy and is a parameter to be fitted. Finally, $k$ is a fitting parameter. It is shown in the Supplementary Information that the value of $B_{50}$ is also equal to the dimensionless free energy of transfer from gas phase to the binding site, $\beta\Delta F_{trans}$. From here, it is trivial to calculate the Gibbs binding free energy as the difference between the transfer free energy, the solvation free energy, $\mu'_{sol}$, and the standard

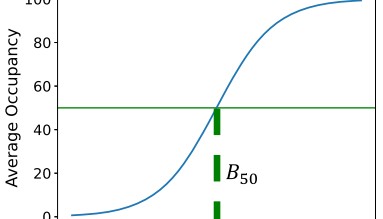

$$B_{eq}(c_L) = \beta\mu'_{sol} + \ln(N_A c_L V_{GCMC})$$

$$c_L = \frac{e^{B_{eq} - \beta\mu'_{sol}}}{N_A V_{GCMC}}$$

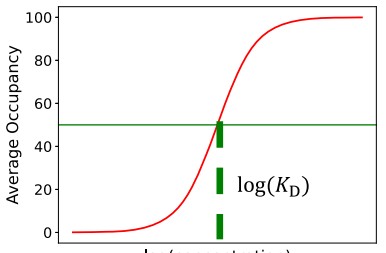

**Fig. 10 | Schematic of GCNCMC titrations.** Simulations are performed at multiple B values, and the final average binding site occupancy of each simulation is plotted such that $B_{50}$ can be calculated using Eq. (15). A simple transformation allows the same data to be plotted on the concentration scale (right) such that $K_D$ can be calculated using Eq. (18).

state correction, $k_B T \ln\left(\frac{V_{\text{GCMC}}}{V^\ominus}\right)$:

$$\Delta G^\ominus = \Delta F_{\text{trans}} - \mu'_{\text{sol}} - k_B T \ln\left(\frac{V_{\text{GCMC}}}{V^\ominus}\right) \qquad (16)$$

The error on $\Delta G^\ominus$, $\sigma_{\Delta G^\ominus}$, is therefore calculated as

$$\sigma_{\Delta G^\ominus} = \sqrt{\sigma^2_{\Delta F_{\text{trans}}} + \sigma^2_{\mu'_{\text{sol}}}} \qquad (17)$$

It follows that a different logistic function can be fitted to the titration data versus concentration, to directly calculate the dissociation constant, $K_D$, and thus $\Delta G^\ominus$:

$$N(\log_{10}(c)) = \frac{1}{1 + \exp[k(\log_{10}(K_D) - \log_{10}(c))]} \qquad (18)$$

where $N(\log_{10}(c))$ is the average number of molecules in the GCMC region as a function of the log of the concentration being simulated, $\log_{10}(c)$ is the log of the simulation concentration, and $k$ and $K_D$ are parameters to be fitted. Figure 10 depicts this protocol graphically and a more in-depth derivation can be found in the Supplementary Information.

**Summary of the GCNCMC protocols.** In summary, the entire protocol involves running regular MD to propagate the system, with GCNCMC moves interspersed. An insertion or deletion move is selected with equal probability. For an insertion move, a "ghost" molecule is randomly placed into the GCMC region while for a deletion move, a fully interacting molecule within the GCMC region is randomly selected. The intermolecular nonbonded interactions of the selected molecule are then scaled appropriately throughout the switch. To avoid numerical instabilities as a result of the nonphysical states sampled during a move, a soft-core Lennard Jones potential is used as described previously[63]. For an insertion move, the Lennard Jones interactions are fully switched on before the electrostatics and vice versa for a deletion move to avoid any naked charges. At the end of the NCMC move, the acceptance test is performed according to Eqs. (6) and (7) and, if the move is accepted, the new state is added to the Markov chain. If the move is rejected, a copy of the previous state is added to the chain and the simulation continues.

**Simulation details**
All simulations were performed using OpenMM 7.4.2 (or OpenMM 8.0 in the case of MUP1). All GCNCMC and free energy calculations, were performed using the *grandlig* python module, a plugin to OpenMM, to set up the custom forces and run these calculations.

Simulations were performed at 298 K and all MD was performed using the Langevin[86,87] BAOAB integrator with a friction coefficient of 1 ps$^{-1}$ and a time step of 2 fs or, for MUP1, 4 fs with hydrogen mass partitioning (Hydrogen mass = 2 Da). Where appropriate, a Monte Carlo barostat is used to maintain a system pressure of 1 bar. The cut-off for nonbonded interactions was 12 Å with a switching function applied at 10 Å for the Lennard-Jones interactions. Particle mesh Ewald (PME) was used to calculate the effect of the long-range electrostatics[88]. Owing to software limitations, the long-range dispersion correction is neglected, as per our previous work[59].

The proteins T4L99A and MUP1 were modeled using the AMBER ff14SB forcefield[89]. The Q4MD-CD[90] forcefield, designed specifically for cyclodextrins, was used for the host $\beta$CD. All simulations use TIP3P[91,92] waters and all ligands are parameterized using GAFF[93] with AM1-BCC charges[94]. Ions, wherever present, were modeled with Joung-Cheatham parameters[95,96].

**Calculation of the excess chemical potential.** The various ligands used in this study are shown in Supplementary Fig. 8, Supplementary

Fig. 11, and Supplementary Fig. 23. A pre-requisite for GCNCMC simulations is to calculate the excess chemical potential, $\mu'_{\text{sol}}$, for the ligand in question. These values were calculated using a basic hydration free energy FEP calculation. The molecule of interest is placed in the center of a 40 × 40 × 40 Å box containing 2135 water molecules. The system is then equilibrated for 3 ns in the NPT ensemble at 298 K. The ligand is then decoupled from the box over 30 lambda values with the first 10 turning off the electrostatics and the final 20 the Lennard-Jones interactions. At each lambda value, the system is equilibrated for 0.5 ns before being run for a further 2 ns with potential energy samples collected every 1000 timesteps. Free energies were calculated using the multistate Bennett acceptance ratio (MBAR) as implemented in *pymbar*[97]. Generally, four repeats per ligand are performed with the mean average taken forward. The standard error of the mean of these four repeats is also calculated and accounted for in all binding free energy calculations. A comparison of the calculated $\mu'_{\text{sol}}$ for ligands with experimental hydration free energies can be found in the Supplementary Information.

**Bulk concentration systems.** In these tests, as the target concentrations (0.5 M Acetone and 0.1 M Pyrimidine) are no longer sufficiently dilute to approximate the excess chemical potential using an infinitely dilute hydration free energy calculation, we require a rigorous parametrization of the excess chemical potential of these species at these specific concentrations. A full discussion on this matter can be found in Section 2 of the Supplementary Information. Using the same protocol as above, we decouple a molecule of acetone or pyrimidine from a box already containing 0.5 M or 0.1 M of acetone or pyrimidine, respectively. Further, in order to fully control the concentration of our test systems, we must also perform GCNCMC moves of the water molecules in the box, and therefore we also parameterize the excess chemical potential of water in these two solutions, although we found that it does not differ from that of bulk water since it is still the dominant species in the solution.

For 0.5 M acetone in water, the calculated $\mu'_{\text{sol}}$ values were −3.25 ± 0.03 and −6.09 ± 0.01 kcal mol$^{-1}$ for acetone and water, respectively. The average volume per acetone and water molecule was 3360 ± 0.9 and 31.5 ± 0.01 Å$^3$. For 0.1 M pyrimidine, the $\mu'_{\text{sol}}$ values were −4.49 ± 0.02 for pyrimidine and −6.09 ± 0.01 kcal mol$^{-1}$ for water. The average volume per pyrimidine and water molecule was 16,312 ± 9 and 30.6 ± 0.01 Å$^3$. The average volume per ligand was calculated by recording the ratio of the number of ligands to box volume throughout a 5 ns NPT simulation at the appropriate concentrations.

The starting points for these tests were equilibrated boxes of pure water containing no other species, and also boxes containing solutions of 1 M acetone and 0.5 M pyrimidine. We then alternate between GCNCMC moves of the ligand and water to control the concentration of the system in the grand canonical ensemble ($\mu$VT). For every 20 ps of MD, one ligand move and three water moves were performed. The switching times for the ligand and water moves were 50 ($n_{\text{pert}} = 499$, $n_{\text{prop}} = 50$) and 10 ($n_{\text{pert}} = 99$, $n_{\text{prop}} = 50$) picoseconds, respectively. The volume of the GCMC region was the volume of the system: 62 nm$^3$.

Eight and four repeats for acetone and pyrimidine from each starting point were performed. The data plotted in the timeseries in Fig. 2 are the mean averages across all 10 repeats at a given move with standard error of the mean shown in the shaded regions. Full details of the simulations performed including the initial concentrations, parameters and final results can be found in Supplementary Table 2.

**Host-guest system.** The coordinates of the host, $\beta$CD, were taken from a review by Mobley et al.[65] and solvated in TIP3P[91] water with an 8 Å buffer. GCNCMC/MD titrations were performed with a switching time of 50 ps ($n_{\text{pert}} = 499$ and $n_{\text{prop}} = 50$). The GCMC region was defined as a

sphere with a 5 Å radius centered in the host cavity midway between two carbon atoms on either side of the host. A total of 22 ligands, from two different studies, with experimental binding affinities were selected for titration and are shown in Supplementary Fig. 23[72,98]. Excess chemical potentials for the ligands were calculated as described above and are reported in the Supplementary Information. Titration $B$ values ($n = 20$) were chosen to loosely surround the experimental dissociation constant. 1700 cycles of GCNCMC/MD were performed at each $B$ value with the first 200 being discarded as equilibration. In these simulations, each cycle consisted of a GCNCMC move attempted for every 1 ps of MD. Each $B$ value was simulated for four repeats. Titration calculations are compared to ABFE calculations using a flat bottom restraint and are described in the Supplementary Information (Section 3.1).

**T4L99A**. The *apo* structure of T4L99A (pdb: 4w51) was protonated according to a pH of 7.0 and missing loops were added where appropriate using PDBFixer[99]. Protein termini were capped using N-methyl and acetyl caps. Each system was then solvated in a box of TIP3P[92] water with a buffer of 12 Å around the protein. NaCl ions were added to neutralize the system and up to a salt concentration of 0.15 M.

In enhanced MSMD simulations looking for the benzene binding site, the protein was solvated in a 0.5 M benzene and water solution. The GCMC sphere was centered on the middle of the protein at the midpoint between the CA atoms of Phe104 and Glu11 with a radius of 26.5 Å to cover the whole protein. The infinitely dilute excess chemical potential of benzene was taken to be −0.68 kcal mol⁻¹ calculated as described above. The average volume per ligand was taken as 3321 Å³ to define a concentration of 0.5 M (Eq. (4)). Using a switching time of 50 ps, we ran six repeats of 700 GCNCMC/MD cycles (1 move per 50 ps of MD) with the first 200 cycles discarded as equilibration, giving a maximum simulation time of 50 ns (25 ns of MD with 25 ns of switches). For a fair comparison, GCNCMC simulations were compared to 50 ns simulations of conventional NPT MD on the same system.

We perform GCNCMC/MD simulations with T4L99A and toluene using the $B$ value which gives a 50% bound occupancy (determined from titrating the $B$ value, see below) as this maximizes the number of insertion/deletion moves that are accepted, maximizing binding and unbinding events. The GCMC sphere was centered on the binding site at the midpoint between the CA atoms of Leu85 and Ala100 with a radius of 8 Å. The Adams value, $B$, was taken to be -7.34 as determined from the titration curves to give 50% ligand occupancy. The dihedral angle between the CA of Arg119 and three toluene atoms was measured at each frame and binned onto a histogram. Dihedral angles between −π to -1.5 and 0 to 1.5 were assigned to binding modes the crystal poses A1 and A2, respectively. The secondary poses, B1/B2 were assigned to angles between -1.5 to 0 and 1.5 to π.

Titrations were performed over 20 B values loosely centered around the experimental binding free energy, although knowledge of the experimental binding affinity is not necessary and the titration could be performed over any B range. B values were calculated using Eq. (3) with a fixed $\mu'_{sol}$ value for each ligand calculated using a basic hydration free energy calculation as described above. Insertions and deletions were performed with a switching time of 150 ps. In each cycle, a GCNCMC move was attempted for every 1 ps of MD. 1700 cycles were performed, with the first 200 discarded as equilibration, giving a maximal production time of 76.5 ns per B value. It should be noted that this protocol is far from optimized. Titration calculations are compared to ABFE calculations which are described in the Supplementary Information (Section 3.1).

**MUP1**. All simulations of MUP1 start from a protein-ligand complex with PDB code 1I06. The crystal ligand was removed but crystallographic waters were retained. The protein was protonated according to a pH of 7.0 using PDBFixer[99]. Missing loops were added where appropriate with PDBFixer and the protein termini were capped using N-methyl and

acetyl caps. The system was then solvated in a box of TIP3P[92] water with a buffer of 12 Å around the protein. NaCl ions were added to neutralize the system and further added to a concentration of 0.15 M.

To find the occluded binding site, GCNCMC simulations (5 repeats) with systems containing 0.5 M of ligands **07, 08**, and **14** (Supplementary Fig. 23) were run as these are the smallest ligands in their series. Simulations were run for 25 ns with 500 GCNCMC moves interspersed every 50 ps for a maximum simulation time of 50 ns. The GCMC region was designed to cover the whole protein anchored to the CA atom of Gly136 with a radius of 22 Å. The simulations were compared to 50 ns conventional MD simulations of the same systems. Insertions and deletions for these simulations were performed with a switching time of 50 ps.

In this study, we perform titration calculations for 14 structurally diverse small molecules binding to MUP1 shown in Supplementary Fig. 23. Titrations were performed over 17 B values between −25 and −12, loosely corresponding to a concentration range of nanomolar to millimolar. In each cycle, a GCNCMC move was attempted for every 1 ps of MD. The GCMC sphere for titration calculations was defined between Phe74 and Leu123 with a radius of 5.5 Å to cover the binding site. In the simulations of MUP1, to avoid wasting computational time at the high and low concentrations, where moves are rarely accepted, the simulations were terminated after 200 consecutive rejected moves. We aim to provide a more rigorous solution for identifying convergence in the future. Titration calculations are compared to ABFE calculations which are described in the Supplementary Information (Section 3.1).

## Data analysis

**Occupancy grids.** To analyze classical and GCNCMC enhanced MSMD simulations we perform a basic grid analysis similar to previously reported MSMD studies[14,30]. After trajectory alignment, a fictitious grid is built in the system with grid voxels spaced 0.5 Å apart. Then, for each frame of the trajectory we loop over all the heavy atoms of all the probes in the system (note: for GCNCMC simulations we only loop over the non-ghost probes). If a probe heavy atom is within 1.6 Å of any voxel, that voxels occupancy for that frame is assigned the number 1. The total occupancy of each voxel is then calculated by summing the number of frames where a probe was present. The final summation is then divided by the total number of frames to provide an 'average occupancy' for each voxel:

$$< O >_{xyz} = \frac{\sum_i^{N_{frames}} O^i_{xyz}}{N_{frames}} \tag{19}$$

where $< O >_{xyz}$ is the average occupancy of a voxel at positions $x$, $y$ and $z$. $O^i_{xyz}$ is the occupancy of a voxel at frame $i$ and $N_{frames}$ is the total number of frames in the simulation.

**Titration curves.** For titration calculations, simulations are performed individually at various values of $B_{eq}$. Four repeats at each $B_{eq}$, and thereby ligand concentration, are performed. The curve described in Eq. (15) is then fitted to each repeat to obtain values of $B_{50}$. Note, in principle occupancy data can fitted on either the B scale (Eq. (15)) or the concentration scale (Eq. (18)). Generally, we find it simpler to calculate $B_{50}$. The mean and standard error of $K_D$, derived from each $B_{50}$ value, over the four repeats are used to calculate $\triangle G^\ominus$ according to Eq. (13). The reported Kendall tau values, indicating the quality of the occupancy data, are calculated from all the occupancy data across the four repeats.

## Reporting summary

Further information on research design is available in the Nature Portfolio Reporting Summary linked to this article.

## Data availability

Input files required to reproduce the results of this paper, and output files containing raw and processed data, are available at: https://github.com/essex-lab/grandlig-paper and Zenodo (https://doi.org/10.5281/zenodo.15310689). Source data are provided with this paper.

## Code availability

The *grand-lig* Python module is available at https://github.com/essex-lab/grand-lig and on Zenodo (https://zenodo.org/records/15261461). Scripts to reproduce the results of this study are found at https://github.com/essex-lab/grandlig-paper.

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

## Acknowledgements

The authors thank the EPSRC, CCP5 (funded under the EPSRC grant EP/V028537/1), Astex Pharmaceuticals, and the University of Southampton for funding. WGP is funded by an EPSRC CASE conversion award with Astex Pharmaceuticals. The authors acknowledge the use of the IRIDIS5 High Performance Computing Facility, and associated support services at the University of Southampton, in the completion of this work. This project made use of time on HPC granted via the UK High-End Computing Consortium for Biomolecular Simulation, HECBioSim (http://hecbiosim.ac.uk), supported by EPSRC (grant nos. EP/R029407/1 and EP/X035603/1).This work made use of the facilities of the N8 Centre of Excellence in Computationally Intensive Research (N8 CIR) provided and funded by the N8 research partnership and EPSRC (Grant No. EP/T022167/1).

## Author contributions

W.G.P., M.L.S., and J.W.E. developed the theoretical formalism and the code. J.W.E. primarily supervised the project with continuous input and discussions between all authors. D.B., M.L.V., M.L.S., and R.D.T. provided industrial supervision. W.G.P. performed all simulations and wrote the manuscript together with J.W.E. and with input from all authors. Correspondence to Jonathan W. Essex.

## Competing interests

The authors declare the following competing financial interest(s): J.W.E. receives funding from UCB and Astex Pharmaceuticals. M.L.S. and R.D.T. are employees of UCB MLV and DB are employees of Astex Pharmaceuticals. Astex Pharmaceuticals part funds W.G.P., where he is now employed.
