## [Transparent Peer Review file · Nature Communications]

Accelerating Fragment Based Drug Discovery using Grand Canonical Nonequilibrium Candidate Monte Carlo

Corresponding Author: Professor Jonathan Essex

Version 0:

Reviewer comments:

Reviewer #1

(Remarks to the Author)

In this manuscript the authors provide a new implementation of fragment-based GCNMC.

They demonstrate how the method is capable of finding occluded fragment binding sites, of sampling multiple fragment binding modes, and of calculating binding affinities for fragment molecules.

As they state in the paper, their application are rather simple systems, which have been successfully studied by many other methods (including additional MC techniques such as in recent studies by the Jorgensen lab).

The methods certainly appear interesting and I believe the paper could be of interest to a more specialized readership, like the one in the JCIM journal, for example. But it lacks wider interest, in my opinion, by the lack of prospective studies and experimental validation. A prospective DD study, with design/validation of novel molecules could make it more interesting for the Nat Com readership.

It is also framing a problem that it really does not exist today in industrial DD. Fragment docking in VS campaigns is already well addressed by a combination of simpler docking techniques followed by high throughput experimental validation. In this sense, I envision GCNMC, a more sophisticated and expensive technique, to be used in H2L phases, where dozens of modification are introduced into a scaffold. In this sense comparison of relative binding free energies in some realistic system would be more desirable.

(Remarks on code availability)

Reviewer #2

(Remarks to the Author)

The application of Grand Canonical Non-equilibrium Candidate Monte Carlo (GCNMC) to fragment-based drug discovery (FBDD) is both innovative and well-executed. The authors successfully extend GCNMC, previously used for water molecule prediction, to the binding of small fragments. This makes the method highly valuable for the early stages of drug discovery, especially in identifying challenging or occluded binding sites. The capability to find binding modes and predict affinities without prior knowledge is impressive. The ability to overcome the time-scale limitations that hinder classical molecular dynamics (MD) simulations is well demonstrated. GCNMC's approach of directly sampling binding/unbinding events in occluded pockets makes it a robust tool, particularly for systems where diffusion limitations prevent effective sampling. Some questions and suggestions:

I have a few questions I want to ask the authors to possibly include in the manuscript:

1. The Val 101 side-chain in T4L99A is known to adopt multiple conformations, especially when binding ligands like paraxylene. These different orientations can influence the local binding pocket and ligand stabilization, yet there is no mention of sampling this flexibility. (p-xylene is not included?) How was the conformational flexibility of Val 101 accounted for in your sampling? Were enhanced sampling techniques used to ensure accurate representation of this side-chain's multiple orientations, particularly in systems like para-xylene binding? Referring to article [23] cited in your study. Although the valine side-chain is slow to sample, it would make it an ideal test case for GCNMC-based side-chain rotational

sampling.

2. The convergence data/plots for the several binding free energies are not explicitly provided. Maybe a distribution of how many data points converged in the given timescale of the method would be able to assess the sufficiency of the sampling and whether the reported results are representative of a fully converged system. This is particularly important in methods like GCNMC, where transitions between states can occur on varying timescales.

3. The paper reports overestimated fragment concentrations at higher levels. Have alternative methods for calculating the excess chemical potential been considered to improve this? Exploring the concentration-dependent corrections to μ^{sol} would be helpful, especially at higher concentrations where current assumptions may break down.

4. Comparison of Timescales (GCNMC vs. FEP) The computational efficiency of GCNMC has not been explicitly compared to Absolute FEP in terms of the timescales required to achieve convergence. How about a small table comparing the time scale to your binding free energy results to FEP with Boresch restraints as well as flat bottom harmonic restraints? An equivalent time scale is also fine. Part of the reason why this is important is because FEP benchmarking campaigns run 3-5 ns/replica as the standard to conclude their methods/workflows are optimal (albeit at the cost of some bad correlations here and there). Adding the faster convergence of GCNMC would not only be encouraging to readers and shine your method in a positive light, it would also show that different systems may need different timescales to stabilize than the standard.

Now onto the suggestions:

1. In Fig. 9, could the authors please include the name of the system(s) in the plot.

2. The authors focus on two protein systems (T4 Lysozyme L99A and MUP1) and a β -cyclodextrin host-guest system, which are excellent starting points. However, for broader applicability, it would be beneficial to test GCNMC on additional, more pharmaceutically relevant targets, particularly those involving larger or more flexible binding sites. The current systems are relatively well-behaved and rigid; extending the method's application to more flexible proteins or systems with conformational variability would further establish its robustness.

(Remarks on code availability)

Reviewer #3

(Remarks to the Author)

Pool et al. have carried out a high-quality study that presents an original approach for computing protein-ligand binding free energies. By using GCNMC to dynamically insert/delete small organic molecules around the vicinity of a protein, they can produce a rich structural and thermodynamic characterization of binding events. This suggests that GCNMC may become a novel CADD approach for early-stage drug discovery. I believe the work will be of broad interest and is suitable for the journal. I also found the SI to be detailed and informative, and the authors should be commended for putting together such a thorough yet accessible manuscript. However, I do have several remarks that I believe should be considered before the manuscript is accepted for publication.

One recurring point made by the authors is that the binding mode of fragments is "tricky owing to unclear electron density" (page 3) and elsewhere. I would like to see citations to the literature providing a few examples. I am aware of one study that contrasted electron densities measured in X-ray diffracted crystals with multiple binding modes seen in MD simulations (Georgiou et al., *J. Mol. Biol.*, 429(16), 2556-2570, 2017), but this was for very low molecular weight fragments (similar to those used in this study). I am not sure that dynamic binding modes are as problematic for more commonly used larger, feature-rich fragments (MW ca. 200–350 g/mol) that are more likely to engage productively with a protein surface through multiple interactions (though I am happy to stand corrected).

The comparison with ABFE methods is perhaps a little unfair. The authors mention that "restraints often require a large degree of user input, and the wrong choice can noticeably affect the simulation" (page 4). However, this issue has been addressed by the development of automated methods to generate ABFE restraint parameters (Heinzmann & Gilson, *Sci. Rep.*, 2021, 11(1), 1116; Haohao et al., *J. Chem. Inf. Model.*, 2021, 61(5), 2116-2123; Clark et al., *J. Chem. Theory Comput.*, 2024, 20(18), 7806-7828), and it is certainly possible today to run large-scale automated ABFE calculations with minimal user input (Wei et al., *J. Chem. Inf. Model.*, 2023, 63(10), 3171-3185). However, the authors' point about weakly bound fragments that alternate between different binding modes is valid, as orientational restraint schemes may show poorer convergence in such cases (though see my earlier point about the likelihood of this being a major concern).

The authors are careful not to oversell the methodology, pointing out remaining challenges such as the need to efficiently sample water and protein degrees of freedom during ligand insertion/deletion. Notwithstanding these issues, a point not discussed in the current study is the scalability of the approach to more complex ligands. Inspection of the SI shows that "the switching times for the ligand and water moves were 50 (npert = 499, nprop = 50) and 10 (npert = 99, nprop = 50) picoseconds, respectively." How were these parameters derived? How does the acceptance rate vary across the ligand datasets studied, and are there correlations with ligand features such as number of atoms, number of rotatable bonds (noting that this is also a function of B)? More details on these aspects would help readers understand whether the GCNMC

protocols used are generally transferable to other problem classes.

Other minor points:

- (Page 1) "lack of molecular obesity" → "low molecular weight"? While I understand that the term "molecular obesity" is used in the field, the suggested wording seems more scientific.
- (Page 3) "is the most accurate and reliable" → "is an accurate and reliable."
- (Page 4) "Various applications of GCMC include sampling buried water molecules in protein-ligand binding regions, to validate crystal water positions, predict favourable water sites, and calculate the free energies of water networks" → I think the SILCS body of work from the Mackerell group (refs 25, 27) should be cited here.
- Section 2.1, Fig. 2 → Please clarify how the equilibration point is detected.
- (Page 9) "in a real-world setting" → "in an industrial R&D environment."
- Section 2.3 → Consider merging Figures 6 and 7 into a single multi-panel figure.
- Section 2.4 → Consider merging Figures 8 and 9 into a single multi-panel figure.
- Fig. 10, third panel → Move the inset so that it does not cover the data points. Here, it appears the ABFE results are systematically noisier. Can the authors comment on this? Is there an intrinsic advantage to GCNMC? It is not clear whether the computational efforts are comparable (though this is not the main point of the study). The methodology used to estimate uncertainties for GCNMC binding free energies (Section 4.3.2) is quite different from that used to estimate the ABFE uncertainties (SI, Section 6). How reproducible are the GCNMC binding free energies between different completely independent repeats of the protocol used here?

(Remarks on code availability)

Code developed following good practices (CMS Cookie cutter template, clear OSS license). I did not attempt to run the code.

Version 1:

Reviewer comments:

Reviewer #1

(Remarks to the Author)

This is clearly a nice paper, no objection. But methods are not new, systems are not new and everything is done retrospectively. The prospective applications will tell if it works in DD.

Today, with better structural target identification, ultra large on demand libraries with AI/MM active learning cycles and DEL libraries and generative modelling, in most targets one find multiple and very active hits in a matter of weeks. Routinely reaching two digit hit rates is accomplished in most virtual screening campaigns, as seen in the recent DDC or Discovery on Target conferences.

In my opinion, it should be published in a more specific journal.

(Remarks on code availability)

Reviewer #2

(Remarks to the Author)

I would like to thank the authors for their thoughtful and detailed responses to my comments. I appreciate the substantial revisions made to the manuscript and supplementary information, particularly the inclusion of additional ligands, convergence analyses, and clarification on concentration-dependent effects. The clarifications provided help strengthen the manuscript and make the methodology more transparent. I offer a few follow-up suggestions and comments below that may help improve clarity and further enhance the manuscript.

1. Thank you for the clarification and for including p-xylene along with other ligands affecting Val111. The added discussion in the SI is appreciated. It may be worth briefly noting whether incorporating side-chain growth into GCNMC proposals is feasible for future applications, or if it's prohibitively expensive or system-dependent.

2. thank you for adding the convergence plots.

3. all comments have been addressed

4. Thank you for the added convergence comparisons. As a future enhancement, and optional to authors, a summary table or visual comparison (e.g., simulation time, force evaluations, or switching time per system) between GCNMC and ABFE would be helpful for readers to quickly evaluate performance trade-offs across systems. This would further highlight where GCNMC may have unique advantages.

(Remarks on code availability)

-

Reviewer #3

(Remarks to the Author)

The authors have satisfactorily addressed comments made on an earlier version of this manuscript. This reviewer also considers the answers to the comments raised by the other reviewers satisfactory. Overall this round of peer-review has increased further the quality of the manuscript and I recommend publication of the work in this journal.

(Remarks on code availability)

REVIEWER COMMENTS

We would like to thank all the reviewers for their carefully considered and constructive feedback. In addition to addressing the points they raised as indicated below, we have taken the opportunity to improve parts of the text which we felt were ambiguous and to correct some issues we identified on further reading. All significant changes are highlighted in the main manuscript and supporting information.

Reviewer #1

(Remarks to the Author):

In this manuscript the authors provide a new implementation of fragment-based GCNMC.

They demonstrate how the method is capable of finding occluded fragment binding sites, of sampling multiple fragment binding modes, and of calculating binding affinities for fragment molecules.

As they state in the paper, their application are rather simple systems, which have been successfully studied by many other methods (including additional MC techniques such as in recent studies by the Jorgensen lab).

The methods certainly appear interesting, and I believe the paper could be of interest to a more specialized readership, like the one in the JCIM journal, for example. But it lacks wider interest, in my opinion, by the lack of prospective studies and experimental validation. A prospective DD study, with design/validation of novel molecules could make it more interesting for the Nat Com readership.

We thank the reviewer for their comment. This manuscript is intended to describe the underlying theory and simulation methodology of the GCNMC approach as applied to fragment binding, and as such we have focused on these aspects. We hope this manuscript will serve as the primary citation for all future studies using this method, by us and the wider community, and by making the software freely available we hope we are facilitating community uptake. While the addition of prospective studies would be beneficial, they would significantly increase the length of the manuscript, and these applications are in a commercial context and would very significantly delay publication for reasons of intellectual property protection.

We would also like to note that other biomolecular simulation methodology papers published in this journal (for example 10.1038/s41467-023-44208-9, 10.1038/s41467-022-28041-0 and 10.1038/s41467-021-23724-6) are similar in structure to our manuscript and do not include full prospective applications.

Finally, we note that the other reviewers are supportive of the work and recognise its applicability to the early stages of drug discovery.

It is also framing a problem that it really does not exist today in industrial DD. Fragment docking in VS campaigns is already well addressed by a combination of simpler docking techniques followed by high throughput experimental validation. In this sense, I envision GCNMC, a more sophisticated and expensive technique, to be used in H2L phases, where dozens of modification are introduced into a scaffold. In this sense comparison of relative binding free energies in some realistic system would be more desirable.

We thank the referee for raising this important point. In response we note that a number of the co-authors of our manuscript have significant experience over many years in the use of fragment-based drug discovery in an industrial context. Their experience is that fragment docking unfortunately is far from a solved problem, with simple docking techniques only predicting the correct binding mode of fragments in the order of 30-40% of the time (10.1021/jm200558u for example). A further complication is that small fragments may present ambiguous electron densities corresponding to a number of possible binding poses (10.1021/jm100677s for example). Additionally, as fragment hits are often extremely weak binders (often >1mM affinity), measuring their affinity can be far from straightforward, and often impossible due to solubility and sensitivity restrictions. The approach we present here predicts the binding mode and the affinity of a fragment simultaneously, and therefore potentially is of great value to both these longstanding challenges in FBDD.

Reviewer #2

(Remarks to the Author):

The application of Grand Canonical Non-equilibrium Candidate Monte Carlo (GCNMC) to fragment-based drug discovery (FBDD) is both innovative and well-executed. The authors successfully extend GCNMC, previously used for water molecule prediction, to the binding of small fragments. This makes the method highly valuable for the early stages of drug discovery, especially in identifying challenging or occluded binding sites. The capability to find binding modes and predict affinities without prior knowledge is impressive. The ability to overcome the time-scale limitations that hinder classical

molecular dynamics (MD) simulations is well demonstrated. GCNCCMC's approach of directly sampling binding/unbinding events in occluded pockets makes it a robust tool, particularly for systems where diffusion limitations prevent effective sampling. Some questions and suggestions:

We would like to thank the reviewer for their positive comments.

I have a few questions I want to ask the authors to possibly include in the manuscript:

1. The Val 101 side-chain in T4L99A is known to adopt multiple conformations, especially when binding ligands like paraxylene. These different orientations can influence the local binding pocket and ligand stabilization, yet there is no mention of sampling this flexibility. (p-xylene is not included?) How was the conformational flexibility of Val 101 accounted for in your sampling? Were enhanced sampling techniques used to ensure accurate representation of this side-chain's multiple orientations, particularly in systems like paraxylene binding? Referring to article [23] cited in your study. Although the valine side-chain is slow to sample, it would make it an ideal test case for GCNCCMC-based side-chain rotational sampling.

We would like to thank the reviewer for this important comment. We believe the reviewer is referring to Val111. Indeed, they are correct to point out that p-xylene, and other T4 ligands, are known to induce a side chain flip of Val111 which has been historically difficult to sample in MD simulations without enhanced protein sampling (see [10.1021/ct700032n](https://doi.org/10.1021/ct700032n), [10.1016/j.jmb.2007.06.002](https://doi.org/10.1016/j.jmb.2007.06.002), [10.1021/acs.jctc.8b01018](https://doi.org/10.1021/acs.jctc.8b01018), for example).

p-xylene was not included in our original dataset, but other ligands which affect the conformation of Val111 were, including o-xylene. To address this deficiency, we have significantly extended the T4L99A dataset to include p-xylene and eight other molecules taken from the study of Mobley et al. ([10.1016/j.jmb.2007.06.002](https://doi.org/10.1016/j.jmb.2007.06.002)). The addition of these new molecules leads to no statistically significant change in the overall performance of the GCNCCMC method. However, owing to the short nonequilibrium switching times used to insert and delete the molecules, these Val111 conformational transitions are often poorly sampled, as in other alchemical methods. To discuss this point, we have added more detail in the main manuscript (page 18) together with a full discussion on the p-xylene/Val111 special case in the SI (Section 2.3.6).

While we could have included the growth of Val111 as part of our GCNCCMC move proposal, we elected not to take this route as it requires *a priori* knowledge of the important

conformational change. Rather we are looking at other enhanced sampling approaches (such as REST2 and GaMD) which may be applied in a more system agnostic way to enhance protein side chain sampling. This substantive piece of work will be the subject of a further publication.

2. The convergence data/plots for the several binding free energies are not explicitly provided. Maybe a distribution of how many data points converged in the given timescale of the method would be able to assess the sufficiency of the sampling and whether the reported results are representative of a fully converged system. This is particularly important in methods like GCNMC, where transitions between states can occur on varying timescales.

We thank the reviewer for this helpful suggestion. To address this point we have added figures illustrating free energy convergence for T4L99A ligands to the supplementary information (pages 27-34). These are data for both GCNMC titrations and ABFE calculations, presented in terms of the percentage of the simulation required to deliver the mean free energy to within one standard error of the final calculated mean. These data are also presented in terms of the number of force evaluations. Overall we believe we demonstrate convincing convergence of the GCNMC data and good agreement with the independent ABFE results.

In addition, the errors now reported on all free energies presented in the manuscript correspond to the standard errors on the mean of a minimum of three repeats.

3. The paper reports overestimated fragment concentrations at higher levels. Have alternative methods for calculating the excess chemical potential been considered to improve this? Exploring the concentration-dependent corrections to μ^{sol} would be helpful, especially at higher concentrations where current assumptions may break down.

We thank the referee for this question, although we believe that we have addressed this point in the manuscript. In the SI we explore the concentration dependence of the excess chemical potential for acetone. There is a significant concentration dependence at higher concentrations, but this dependence is only marginally greater than the random errors on the calculated values.

For the ligand binding studies, the ligand concentrations are sufficiently low such that it is acceptable to use the infinitely dilute excess chemical potential, as in regular free energy calculations.

In the case of the fragment concentration study, the target ligand concentrations are high (0.5 M and 0.1 M for acetone and pyrimidine, respectively), and as such we used the chemical potentials appropriate for these specific target concentration i.e. not the infinitely

dilute values. This results in a concentration slightly too high for acetone (0.55 ± 0.02 M) while the 0.1 M target value for pyrimidine was well reproduced (0.10 ± 0.01 M). In the SI we illustrate the consequences of using the infinitely dilute excess chemical potential for acetone which, despite being only 0.08 kcal mol⁻¹ more positive (-3.25 kcal mol⁻¹ at 0.5 M versus -3.17 kcal mol⁻¹ at infinite dilution), leads to a concentration (approximately 0.7 M) significantly above the target concentration of 0.5 M. As we have previously reported in our water studies of bulk water density (10.1021/acs.jcim.0c00648), small differences in excess chemical potential can have significant effects.

We hope these details address the point raised by the reviewer, but we are happy to add additional detail and calculations if requested.

4. Comparison of Timescales (GCNMC vs. FEP) The computational efficiency of GCNMC has not been explicitly compared to Absolute FEP in terms of the timescales required to achieve convergence. How about a small table comparing the time scale to your binding free energy results to FEP with Boresch restraints as well as flat bottom harmonic restraints? An equivalent time scale is also fine. Part of the reason why this is important is because FEP benchmarking campaigns run 3-5 ns/replica as the standard to conclude their methods/workflows are optimal (albeit at the cost of some bad correlations here and there). Adding the faster convergence of GCNMC would not only be encouraging to readers and shine your method in a positive light, it would also show that different systems may need different timescales to stabilize than the standard.

We thank the reviewer for raising this important point. As indicated in our response to comment 2 above, we have added convergence plots for ABFE and GCNMC simulations for the T4L99A system to the SI, with associated discussion. We find that ABFE simulations are generally more efficient in terms free energy convergence than GCNMC. However, we note that we have not attempted to optimise either protocol, and that ABFE requires knowledge of the available binding poses, whereas these arise naturally from the GCNMC simulations.

Now onto the suggestions:

1. In Fig. 9, could the authors please include the name of the system(s) in the plot.

This has been done. Following the requested merging of Figures 8 and 9 by reviewer 3, this Figure is now Figure 8.

2. The authors focus on two protein systems (T4 Lysozyme L99A and MUP1) and a β -cyclodextrin host-guest system, which are excellent starting points. However, for broader applicability, it would be beneficial to test GCNMC on additional, more pharmaceutically

relevant targets, particularly those involving larger or more flexible binding sites. The current systems are relatively well-behaved and rigid; extending the method's application to more flexible proteins or systems with conformational variability would further establish its robustness.

We thank the reviewer for this important suggestion. Indeed, since performing the work reported in this manuscript, we have used GCNMC to augment the sampling in mixed-solvent molecular dynamics (MSMD) of eight new protein-ligand systems, six of which have occluded binding pockets. We find that the addition of GCNMC moves results in significant improvements over conventional molecular dynamics, with true ligand binding sites being identified and occupied more quickly and with converged populations. There are also far fewer false positives. A manuscript reporting these results is in the final stages of preparation and will be submitted shortly. We are very happy to share the latest draft of that manuscript with the reviewer if that would be welcome. We are also currently exploring the combination of other enhanced sampling methods with GCNMC of fragments, including GCMC of waters, and GaMD and REST2 of protein sidechains. Each of these studies is a significant piece of research and we are proposing that they be published separately.

Reviewer #3

(Remarks to the Author):

Poole et al. have carried out a high-quality study that presents an original approach for computing protein-ligand binding free energies. By using GCNMC to dynamically insert/delete small organic molecules around the vicinity of a protein, they can produce a rich structural and thermodynamic characterization of binding events. This suggests that GCNMC may become a novel CADD approach for early-stage drug discovery. I believe the work will be of broad interest and is suitable for the journal. I also found the SI to be detailed and informative, and the authors should be commended for putting together such a thorough yet accessible manuscript. However, I do have several remarks that I believe should be considered before the manuscript is accepted for publication.

One recurring point made by the authors is that the binding mode of fragments is "tricky owing to unclear electron density" (page 3) and elsewhere. I would like to see citations to the literature providing a few examples. I am aware of one study that contrasted electron densities measured in X-ray diffracted crystals with multiple binding modes seen in MD simulations (Georgiou et al., J. Mol. Biol., 429(16), 2556-2570, 2017), but this was for very

low molecular weight fragments (similar to those used in this study). I am not sure that dynamic binding modes are as problematic for more commonly used larger, feature-rich fragments (MW ca. 200–350 g/mol) that are more likely to engage productively with a protein surface through multiple interactions (though I am happy to stand corrected).

We would like to thank the reviewer for this helpful comment. We have now added relevant supporting citations to the statement in the manuscript. In addition, we note that some companies are moving to fragment screens using very small molecules, such as MiniFragments (10.1016/j.drudis.2019.03.009), which are likely to be more problematic in terms of resolving ambiguous electron density.

The comparison with ABFE methods is perhaps a little unfair. The authors mention that "restraints often require a large degree of user input, and the wrong choice can noticeably affect the simulation" (page 4). However, this issue has been addressed by the development of automated methods to generate ABFE restraint parameters (Heinzelmann & Gilson, *Sci. Rep.*, 2021, 11(1), 1116; Haohao et al., *J. Chem. Inf. Model.*, 2021, 61(5), 2116-2123; Clark et al., *J. Chem. Theory Comput.*, 2024, 20(18), 7806-7828), and it is certainly possible today to run large-scale automated ABFE calculations with minimal user input (Wei et al., *J. Chem. Inf. Model.*, 2023, 63(10), 3171-3185). However, the authors' point about weakly bound fragments that alternate between different binding modes is valid, as orientational restraint schemes may show poorer convergence in such cases (though see my earlier point about the likelihood of this being a major concern).

We thank the reviewer for raising this important point. We have amended the manuscript on page 4 to emphasise the role of automated procedures for ABFE restraint identification, including relevant citations. We have also noted that problems may nevertheless arise with small fragments. We hope this response is sufficient to satisfy the reviewer's concern.

The authors are careful not to oversell the methodology, pointing out remaining challenges such as the need to efficiently sample water and protein degrees of freedom during ligand insertion/deletion. Notwithstanding these issues, a point not discussed in the current study is the scalability of the approach to more complex ligands. Inspection of the SI shows that "the switching times for the ligand and water moves were 50 (npert = 499, nprop = 50) and 10 (npert = 99, nprop = 50) picoseconds, respectively." How were these parameters derived? How does the acceptance rate vary across the ligand datasets studied, and are there correlations with ligand features such as number of atoms, number of rotatable bonds (noting that this is also a function of B)? More details on these aspects would help

readers understand whether the GCNCCMC protocols used are generally transferable to other problem classes.

We thank the reviewer for raising this interesting point.

The particular GCNCCMC protocol adopted was based on test calculations on the host guest system and from our previous experience on the development of GCNCCMC for water simulations (10.1021/acs.jctc.2c00823). Optimising the protocol for water required a significant amount of computer power, and performing the equivalent optimisation on each ligand here is arguably impractical. We have however, investigated the GCNCCMC acceptance probabilities as a function of heavy atom count, presenting the data in the supplementary information (Section 2.3.5). We find, unsurprisingly, that the acceptance rates decrease with ligand size, although the acceptance rates for the largest ligands are still acceptable at 0.5-1.5%, and are, of course, system and B value dependent. We also note that many of these rejected moves are associated with the ligand leaving the GCMC region (breaking the condition of detailed balance). Thus we accept that while there is scope for further protocol optimisation, the protocol used here yields acceptable acceptance rates, even for the largest ligands. With further data across more systems it will become possible to see whether a heuristic could be developed to better optimise the GCNCCMC protocol for fragments, and this is something we will look to do as part of future work.

Other minor points:

- (Page 1) "lack of molecular obesity" → "low molecular weight"? While I understand that the term "molecular obesity" is used in the field, the suggested wording seems more scientific.

We have changed the text as requested.

- (Page 3) "is the most accurate and reliable" → "is an accurate and reliable."

We have changed the text as requested.

- (Page 4) "Various applications of GCMC include sampling buried water molecules in protein-ligand binding regions, to validate crystal water positions, predict favourable water sites, and calculate the free energies of water networks" → I think the SILCS body of work from the Mackerell group (refs 25, 27) should be cited here.

We have added the references as requested.

- Section 2.1, Fig. 2 → Please clarify how the equilibration point is detected.

We have added detail to the figure caption to explain the process.

- (Page 9) "in a real-world setting" → "in an industrial R&D environment."

We have changed the text as requested.

- Section 2.3 → Consider merging Figures 6 and 7 into a single multi-panel figure.

We have decided against doing this because in our opinion it would make the new figure unbalanced in terms of layout. If the reviewer feels this is important, we are very happy to revisit this point.

- Section 2.4 → Consider merging Figures 8 and 9 into a single multi-panel figure.

We have merged the figures as requested.

- Fig. 10, third panel → Move the inset so that it does not cover the data points. Here, it appears the ABFE results are systematically noisier. Can the authors comment on this? Is there an intrinsic advantage to GCNMC? It is not clear whether the computational efforts are comparable (though this is not the main point of the study). The methodology used to estimate uncertainties for GCNMC binding free energies (Section 4.3.2) is quite different from that used to estimate the ABFE uncertainties (SI, Section 6). How reproducible are the GCNMC binding free energies between different completely independent repeats of the protocol used here?

We thank the reviewer for drawing this point to our attention. We have moved the inset to reveal the data point. To address the question regarding our errors, we have changed the error estimate procedure to make it consistent between ABFE and GCNMC – namely the standard error on the mean of multiple independent repeats. Previously we were using bootstrapped estimates for GCNMC which arguably yielded unreasonably small error estimates. We have also added a detailed discussion regarding free energy convergence to the supplementary information following the request of reviewer 2. We find that the GCNMC and ABFE errors are now broadly comparable, although as noted above, we have not attempted to optimise either protocol. GCNMC does offer the advantage over ABFE in that it finds binding modes and calculates affinity from a single series of simulations, without the need to invoke restraints or symmetry corrections.

Reviewer #3 (Remarks on code availability):

Code developed following good practices (CMS Cookie cutter template, clear OSS license).
I did not attempt to run the code.

REVIEWER COMMENTS – Final Submission

Reviewer #1

(Remarks to the Author):

This is clearly a nice paper, no objection. But methods are not new, systems are not new and everything is done retrospectively. The prospective applications will tell if it works in DD.

Today, with better structural target identification, ultra large on demand libraries with AI/MM active learning cycles and DEL libraries and generative modelling, in most targets one find multiple and very active hits in a matter of weeks. Routinely reaching two digit hit rates is accomplished in most virtual screening campaigns, as seen in the recent DDC or Discovery on Target conferences.

In my opinion, it should be published in a more specific journal.

Reviewer #2

(Remarks to the Author):

I would like to thank the authors for their thoughtful and detailed responses to my comments. I appreciate the substantial revisions made to the manuscript and supplementary information, particularly the inclusion of additional ligands, convergence analyses, and clarification on concentration-dependent effects. The clarifications provided help strengthen the manuscript and make the methodology more transparent. I offer a few follow-up suggestions and comments below that may help improve clarity and further enhance the manuscript.

1. Thank you for the clarification and for including p-xylene along with other ligands affecting Val111. The added discussion in the SI is appreciated. It may be worth briefly noting whether incorporating side-chain growth into GCNMC proposals is feasible for future applications, or if it's prohibitively expensive or system-dependent.

There is a comment on page 18, and in the SI, discussing the possibilities of coupling sidechain dihedral sampling. In general, we believe it to be feasible, however some methods require knowledge of the rotamer states *a priori* and major code changes for our

implementation would be required. We have also commented briefly on alternate methods.

2. thank you for adding the convergence plots.

3. all comments have been addressed

4. Thank you for the added convergence comparisons. As a future enhancement, and optional to authors, a summary table or visual comparison (e.g., simulation time, force evaluations, or switching time per system) between GCNMC and ABFE would be helpful for readers to quickly evaluate performance trade-offs across systems. This would further highlight where GCNMC may have unique advantages.

A full data table detailing the convergence for all T4L99A simulations for both GCNMC titrations and ABFE as a function of simulation time is available on the github repo and in the source data file. In addition, the number of force evaluations required to achieve convergence for each T4L99A ligand is available in the source data file.

Reviewer #3

(Remarks to the Author):

The authors have satisfactorily addressed comments made on an earlier version of this manuscript. This reviewer also considers the answers to the comments raised by the other reviewers satisfactory. Overall, this round of peer-review has increased further the quality of the manuscript and I recommend publication of the work in this journal.